# FEVERLESS: FAST AND SECURE VERTICAL FEDERATED LEARNING BASED ON XGBOOST FOR DECENTRALIZED LABELS

## ABSTRACT

Vertical Federated Learning (VFL) enables multiple clients to collaboratively train a global model over vertically partitioned data without revealing private local information. Tree-based models, like XGBoost and LightGBM, have been widely used in VFL to enhance the interpretation and efficiency of training. However, there is a fundamental lack of research on how to conduct VFL securely over distributed labels. This work is the first to fill this gap by designing a novel protocol, called FEVERLESS, based on XGBoost. FEVERLESS leverages secure aggregation via information masking technique and global differential privacy provided by a fairly and randomly selected noise leader to prevent private information from being leaked in the training process. Furthermore, it provides label and data privacy against honest-but-curious adversary even in the case of collusion of $n - 2$ out of $n$ clients. We present a comprehensive security and efficiency analysis for our design, and the empirical results from our experiments demonstrate that FEVERLESS is fast and secure. In particular, it outperforms the solution based on additive homomorphic encryption in runtime cost and provides better accuracy than the local differential privacy approach[1].

## 1 INTRODUCTION

Traditional centralized deep learning models, demanding to collect a considerable amount of clients' data to maintain high accuracy, to some degree, may increase the risk of data breaches. Data may not be easily shared among different entities due to privacy regulations and policies. To tackle this "Data Island" problem (Yang et al. (2019a)), Google proposed Federated Learning (FL) (McMahan et al. (2017)) to allow multiple clients to train a global model without sharing private data. The basic paradigm of FL is that all clients train local models with their own data, and then the information of local models, e.g., gradients, may be exchanged to produce a global model.

Based on different types of data partition (Yang et al. (2019a)), FL can be mainly categorized into Horizontal Federated Learning (HFL) and Vertical Federated Learning (VFL). The former focuses on training with horizontally partitioned data where clients share the same feature space but differing in data index set. Several research works (Shokri & Shmatikov (2015); Orekondy et al. (2019); Geiping et al. (2020); Li & Han (2019)) have found that training data of HFL is still at high risk of leakage although private data is kept locally. Other studies (Phong et al. (2018); Truex et al. (2019); Xu et al. (2019); Zhang et al. (2020); Zhu et al. (2020)) have been dedicated to enhancing the security of HFL. On the contrary, VFL is mainly applied in the scenario of training with vertically partitioned data (Wu et al. (2020); Cheng et al. (2021)) where clients share the same data index set but differing in feature space. In this paper, our principal focus is to achieve privacy-preserving training on VFL.

To best of our knowledge, many existing studies (Hardy et al. (2017); Nock et al. (2018); Liu et al. (2020); Yang et al. (2019b); Cheng et al. (2021); Chen & Guestrin (2016); Wu et al. (2020)) have proposed innovative approaches to prevent private information breaches in the context of VFL. Specifically, (Hardy et al. (2017)) introduced encryption-based privacy-preserving logistic regression to safeguard the information of data indexes. (Nock et al. (2018)) gave a comprehensive discussion on

---

[1]Code is available at: https://github.com/feverless111/vfl

the impact of ID resolution. (Yang et al. (2019b)) introduced a scheme without using a coordinator for a limited number of clients. Recently, (Liu et al. (2020)) proposed an asymmetrically VFL scheme for logistic regression tackling privacy concerns on ID alignment.

Unlike the training models used in the aforementioned works, XGBoost (Chen & Guestrin (2016)), which is one of the most popular models applied in VFL, can provide better interpretation, easier parameter tuning, and faster execution than deep learning in tabular data training (Goodfellow et al. (2016); LeCun et al. (2015)). These practical features and advantages draw academia and industry's attention to the research on XGBoost, especially in the privacy-preserving context. (Wu et al. (2020)) introduced an approach for tree-based model training through a hybrid method composing homomorphic encryption and secure Multi-Party Computation (MPC) (Goldreich (1998); Bonawitz et al. (2017)). After that, (Cheng et al. (2021)) proposed a similar system to train XGBoost (Chen & Guestrin (2016)) securely over vertically partitioned data by using Additively Homomorphic Encryption (AHE). By applying Differential Privacy (DP) (Dwork (2008)), (Tian et al. (2020)) designed a VFL system to train GBDT without the need of encryption/decryption.

However, most of the above solutions based on AHE and MPC do not scale well in terms of efficiency on training XGBoost. Beyond that, all the existing schemes basically assume that training labels are managed and processed by a *sole* client. In practice, *a VFL scheme supporting distributed labels is necessary*. For instance, multiple hospitals, clinics and health centers currently may be set to COVID-19 test spots and aim to train a model, e.g., XGBoost, to predict with good interpretation if citizens (living in various locations) are infected based on their health records and symptoms. In this context, the labels (i.e., the test results) are likely distributed among different health authorities - even targeting to the same group of patients, and feature space is vertically portioned. For example, a cardiac hospital only maintains heart data for the patients, while a psychiatric center holds the mental records, in which both authorities may collect and manage each of its registered patient's label locally. Another common scenario could be in the financial sector where multiple bank branches and e-commerce companies prefer to build a global model to predict if their customers may pay some service (e.g., car loan) on time. The banks have part of features about the customers (e.g., account balance, funding in-and-out records), while the companies may obtain other features (e.g., payment preference). Since the customers may get the same service, e.g., loan, from different institutions, it is clear that labels must be distributed rather than centralized. In addition to efficiency and functionality aspects, one may also consider capturing stronger security for VFL. Training an XGBoost usually should involve the computation of first and second-order derivatives of the loss function (note gradients and hessians contain labels' information), and the aggregation of them is required in each round. In the context where the labels are held by different clients, if the gradients and hessians are transmitted in the form of plaintexts and the summations of them are known to an aggregator (whom could be one of the clients engages in training), inference and differential attacks (Appendix C) will be easily conducted by the aggregator, resulting in information leakage.

To tackle these problems, we propose a fast and secure VFL protocol, FEVERLESS, to train XGBoost (Appendix B.1) on distributed labels without disclosing both feature and label information. In our design, the privacy protection is guaranteed by secure aggregation (based on a masking scheme) and Global Differential Privacy (GDP) (Appendix B.6). We leverage masking instead of heavy-cost multiparty computation and we guarantee a "perfect secrecy" level for the masked data. In GDP, we use Verifiable Random Function (VRF) (Appendix B.5) to select a noise leader per round (who cannot be predicted and pre-compromised in advance) to aggregate noise from "selected" clients, which significantly maintains model accuracy.

**Our contributions** can be summarized as follows.
(1) We define VFL in a more practical scenario where training labels are distributed over multiple clients. Beyond that, we develop FEVERLESS to train XGBoost securely and efficiently with the elegant combination of secure aggregation technique (based on Diffie-Hellman (DH) key exchange (Appendix B.2) and Key Derivation Function (KDF) (Appendix B.4)) and GDP.
(2) We give a comprehensive security analysis to demonstrate that FEVERLESS is able to safeguard labels and features privacy in the semi-honest setting, but also maintain the robustness even for the case where $n - 2$ out of $n$ clients commit collusion.
(3) We implement FEVERLESS and perform training time and accuracy evaluation on different real-world datasets. The empirical results show that FEVERLESS can maintain efficiency and accuracy simultaneously, and its performance is comparable to the baseline - a "pure" XGBoost without using any encryption and differential privacy. Specifically, training the credit card and bank marketing

datasets just takes $1\%$ and $6.5\%$ more runtime than the baseline and meanwhile, the accuracy is only lower than that of the baseline by $0.9\%$ and $3.21\%$, respectively[2].

## 2 PROBLEM FORMULATION

### 2.1 SYSTEM MODEL

Before proceeding, we give some assumptions on our model. We suppose that a private set intersection (Kolesnikov et al. (2017); Pinkas et al. (2014)) has been used to align data IDs before the training starts, so that each client shares the same data index space $\mathcal{I}$. But the names of features are not allowed to share among clients. As for the information of label distribution (indexes indicating a label belongs to which client, e.g., the label of $i$-th data instance is held by client $A$), we will consider the following conditions: (1) this information is revealed to the public in advance; or (2) the information is not allowed to publish but the training can still be accomplished (with extra cost).

We also consider that the training is conducted on a dataset with $m$ samples composing with feature space $\mathcal{X} = \{x_1, \cdots, x_m\}$, each containing $f$ features, and label set $\mathcal{Y} = \{y_1, \cdots, y_m\}$. Besides, features $\{X_j^{(c)} \mid j \in \{1, \cdots, f\}\}$ and labels $\{y_i^{(c)} \mid i \in \{1, \cdots, m\}\}$ are held among $n$ clients where each client has at least one feature and one label. $X_j^{(c)}$ and $y_i^{(c)}$ refer to $j$-th feature and $i$-th label owned by $c$-th client, respectively.

Considering a practical scenario wherein training labels are distributed among clients, we propose a new variant of VFL, named VFL over Distributed Labels (DL-VFL). The concrete definition is given as follows.

**Definition 1** (DL-VFL). *Given a training set with $m$ data samples consisting of feature space $\mathcal{X}$, label space $\mathcal{Y}$, index space $\mathcal{I}$ and clients set $\mathcal{C}$, we have:*

$$\mathcal{X}^c \cap \mathcal{X}^{c'} = \emptyset, \mathcal{Y}^c \cap \mathcal{Y}^{c'} = \emptyset, \mathcal{I}^c = \mathcal{I}^{c'}, \forall c, c' \in \mathcal{C}, c \neq c'. \tag{1}$$

A client $c$ participating DL-VFL shares the same sample ID space $\mathcal{I}$ with the corresponding labels, where a single label belongs to only one client. And different clients hold the subset of $\mathcal{X}$ sampled from feature space. To achieve privacy-preserving XGBoost training, we further define two roles.

**Definition 2** (Source client). *A source client with split candidates wants to compute the corresponding $L_{split}$ based on Eq.(4). But some labels are missing so that $\sum g_i$ and $\sum h_i$ are unable to derive.*

For the case that a source client does not hold all labels in the current split candidates, we propose a solution based on secure aggregation and global differential privacy to help the source client to compute $L_{split}$ while safeguarding other clients' privacy. We consider the two conditions regarding if label distribution is publicly known. We find that if we keep label distribution hidden, we will take extra communication overhead to perform training. The detailed explanation is given in Appendix F. Note each client may have a chance to act as a source client because all the labels are distributed, where the *source client* leads the $L_{split}$ computation, and *clients* provide missing label values to the source client.

To achieve GDP, we define noise leader who is selected fairly and randomly from all clients (except for the source client) - preventing clients from being compromised beforehand.

**Definition 3** (Noise leader). *By using VRF, a noise leader is responsible for generating the maximum leader score, aggregating differentially private noise from a portion of clients and adding the noise to the gradients and hessians.*

Note we summary the main notations in Table 1 (see Appendix A).

### 2.2 THREAT MODEL

We mainly consider potential threats incurred by participating clients and the outside adversaries. We assume that all clients are *honest-but-curious*, which means they strictly follow designed algo-

---

[2]For banknote authentication dataset, FEVERLESS takes $13.96\%$ more training time than the baseline, and the accuracy is $30.4\%$ lower. This is because the model is trained by a small-scale dataset, so that the robustness is seriously affected by noise.

rithms but try to infer private information of other clients from the received messages. Besides, we also consider up to $n-2$ clients' collusion to conduct attacks, and at least one non-colluded client adds noise per round. Through authenticated channels, DH key exchange can be securely executed among clients. Other messages are transmitted by public channels, and outside attackers can eavesdrop on these channels and try to reveal information about clients during the whole DL-VFL process. Note this paper mainly focuses on solving privacy issues in training DL-VFL based on XGBoost. Thus, other attacks, like data poisoning and backdoor attacks deteriorating model performance, are orthogonal to our problem.

## 3 A PRACTICAL PRIVACY-PRESERVING PROTOCOL

### 3.1 FEVERLESS PROTOCOL DESCRIPTION

To prevent a source client from knowing gradients and hessians sent by other clients, one may directly use MPC (Damgård et al. (2012)) based on AHE (Paillier (1999); Wu et al. (2020)). But this method yields expensive computation cost. Getting rid of the complex mechanism like MPC, we leverage secure aggregation protocol via masking scheme based on DH key exchange(Bonawitz et al. (2017); Ács & Castelluccia (2011); Tian et al. (2020)). By further using KDF and Hash Function (see Appendix B.3&B.4), our masking (for gradients and hessians) can be derived without exchanging keys per training round. Our approach significantly reduces the communication cost but still maintains the robustness up to $n-2$ colluded clients. Meanwhile, the secure aggregation can provide "perfect secrecy" for broadcast messages. After receiving the broadcast messages, the masking will be canceled out at the source client side. But only using the masking is unable to defend against differential attacks. One may consider using Local Differential Privacy (LDP) (Kairouz et al. (2014)) to make sure that each client may add noise to per send-out message, barely consuming any extra computation cost. The accumulated noise, from all clients, may seriously affect the model accuracy. To tackle this problem, we use a GDP (Wei et al. (2020)) approach with noise leader selection. A hybrid method is finally formed based on masking scheme and GDP, so that per client's sensitive information can be protected by the "masks" and the aggregated values are secured by the noise which is injected by the chosen clients.

We briefly introduce our design, and the detailed algorithms and more explanations are given in Appendix D. Assume each client $c \in [1, n]$ generates respective secret key $sk_c$ and computes gradients $g_i^{(c)}$ and hessians $h_i^{(c)}$ locally, where $\{i \mid y_i \in \mathcal{Y}^c\}$. FEVERLESS works as follows.

1. *Broadcast missing indexes.* The source client broadcasts the $mID$s= $\{i \mid y_i \notin \mathcal{Y}^c\}$.

2. *Key exchange computation.* Each client $c$ computes public key $pk_c = g^{sk_c}$ using secret keys $sk_c$, sends $pk_c$ to other clients and computes the corresponding shared keys[3] $\{S_{c,c'} = pk_{c'}^{sk_c} = g^{sk_c sk_{c'}} \mid c, c' \in \mathcal{C}, c \neq c'\}$ based on secret key $sk_c$ received public keys $\{pk_{c'} \mid c' \in \mathcal{C}\}$.

3. *Data masking.* Each client $c$ runs the masking generation algorithm to compute the maskings for protecting gradients and hessians. Specifically, based on KDF, clients' indexes and the number of queries, the masking generation algorithm is conducted by $\text{mask}_g^{(c)} \leftarrow \sum_{c \neq c'} \frac{|c-c'|}{c-c'} \cdot \left( \text{H}(S_{c,c'} \| 0 \| \text{query}) \right)$, $\text{mask}_h^{(c)} \leftarrow \sum_{c \neq c'} \frac{|c-c'|}{c-c'} \cdot \left( \text{H}(S_{c,c'} \| 1 \| \text{query}) \right)$ [4]. Then the masked gradients $G^{(c)}$ and hessians $H^{(c)}$ are generated by $G^{(c)} = \sum_{i \in mIDs} g_i^{(c)} + \text{mask}_g^{(c)} - r_g^{(c)}$, $H^{(c)} = \sum_{i \in mIDs} h_i^{(c)} + \text{mask}_h^{(c)} - r_h^{(c)}$.

4. *Noise leader selection.* Each client generates the selection score $selec_c$ using the VRF, $\text{H}(\text{SIGN}_{sk_c}(\text{count}, \text{mIDs}, \text{r}))$, and broadcasts it, where $\text{count}$ is the number of times clients conduct VRF, $\text{r}$ is a fresh random number, and $\text{SIGN}$ is the signature scheme (see Appendix B.5 for more details). The client with maximum score will be the noise leader. For ease of understanding, we assume client $n$ with the largest selection score $select_n^{max}$ is the leader, in Figure 1.

---

[3]Shared keys are only generated once, and the KDF is used to generate the remaining maskings.

[4]For purpose of simplicity, we omit modular computations. The complete calculation processes are elaborated on Algorithm 3-5.

5. *Noise injection.* a) Noise leader selects $k$ clients adding noise. For the details of the selection, please see Algorithm 5 in Appendix D. b) The selected clients send $\{\widetilde{n_g^{(c)}} = N(0, \Delta_g^2\sigma^2) + r_g^{(c)}, \widetilde{n_h^{(c)}} = N(0, \Delta_h^2\sigma^2) + r_h^{(c)} | c \in k\}$ to noise leader, in which the $r_g^{(c)}$ and $r_h^{(c)}$ are two random values to mask noise. c) The leader aggregates the noise: $\widetilde{N_g} = k \cdot N(0, \Delta_g^2\sigma^2) + R_g$ and $\widetilde{N_h} = k \cdot N(0, \Delta_h^2\sigma^2) + R_h$, and further adds them to $G^{(n)}$ and $H^{(n)}$, respectively.

6. *Aggregation and computation.* All clients send the masked values to the source client. The source client computes $\sum_{c=1}^{n} G^{(c)} + k \cdot N(0, \Delta_g^2\sigma^2)$, $\sum_{c=1}^{n} H^{(c)} + k \cdot N(0, \Delta_h^2\sigma^2)$ and $L_{split}$.

7. *Final update.* The source client with maximum $L_{split}$ updates model following XGBoost (Chen & Guestrin (2016)) and broadcasts the updated model and data indexes in child nodes as step 8.

Figure 1 gives an overview of FEVERLESS. Note this process can be conducted iteratively. For simplicity, the core calculation processes are shown here, and more details are in Appendix D.

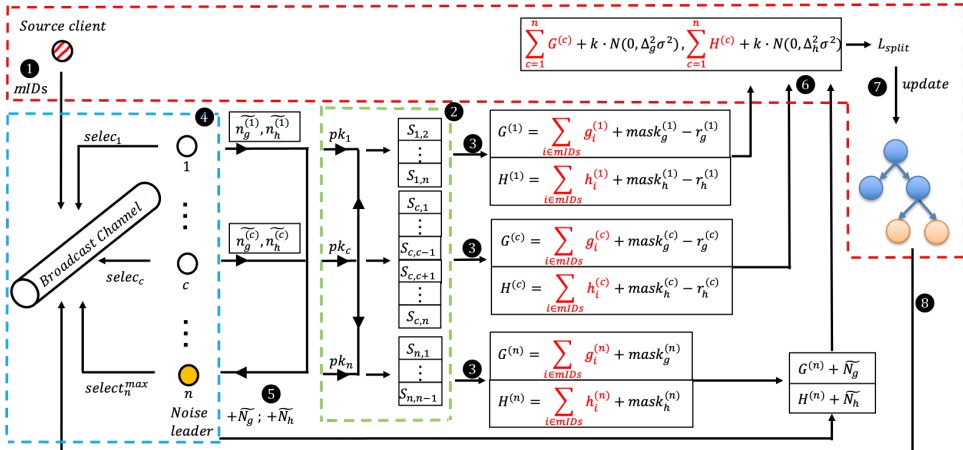

Figure 1: Overview of FEVERLESS. ---- : Source client broadcasts missing $ID$s, aggregates gradients and hessians securely, updates model and broadcasts nodes $ID$s. ---- : DH key exchange and maskings generation. ---- : Noise leader selection. ❶Broadcast missing indexes. ❷Key exchange computation. ❸Data masking. ❹Noise leader selection. ❺Global noise injection. ❻Aggregation and computation. ❼❽ Final update and broadcast updated model. Note sensitive data are in red. The maskings in ❸ protect data from source client, and the noise in aggregated gradients and hessians prevents source client from conducting differential attack.

## 3.2 THEORETICAL ANALYSIS

**Computation cost:** We use $B$ and $d$ to denote the number of buckets and the maximum depth respectively, and $f^{(c)}$ here represents the number of features held by a client $c$. For each client $c$, the computation cost can be divided into 4 parts: (1) Performing at most $f^{(c)} \cdot B \cdot NT \cdot (2^d - 1)$ times computation of $L_{split}$ and $w$, taking $O(f^{(c)} \cdot B \cdot NT \cdot 2^d)$ time; (2) Creating $n - 1$ shared keys and 1 public key, which is $O(n)$; (3) Conducting $O(f^{(c)} \cdot B \cdot NT \cdot 2^d)$ time to compute VRF outputs, select noise leader and generate noise; (4) Generating $2f^{(c)} \cdot B \cdot NT \cdot (2^d - 1)$ maskings, which takes $O(f^{(c)} \cdot B \cdot NT \cdot 2^d \cdot n)$ time. Overall, each client's computation complexity is $O(f^{(c)} \cdot B \cdot NT \cdot 2^d \cdot n)$.

**Communication cost:** Each client's communication cost can be calculated as (1) Broadcasting at most $f^{(c)} \cdot B \cdot NT \cdot (2^d - 1)$ times of missing indexes $mID$; (2) Broadcasting 1 public key and receiving $n - 1$ public keys from other clients; (3) Broadcasting 1 leader selection score and sending noise to noise leader at most $f^{(c)} \cdot B \cdot NT \cdot (2^d - 1)$ times; (4) Sending source client 2 masked gradients and hessians of size $2\lceil log_2 N \rceil$. Therefore the overall communication cost is $f^{(c)} \cdot B \cdot NT \cdot (2^d - 1) \cdot (\|mID\| \cdot \alpha_I + \alpha_L + \alpha_N + n \cdot \alpha_K 2\lceil log_2 N \rceil)$, where $\alpha_I, \alpha_L, \alpha_N$ and

$\alpha_K$ refer to the number of bits of index, leader selection score, noise and public keys, respectively. Thus, we have the communication complexity $O(f^{(c)} \cdot B \cdot NT \cdot 2^d)$.

## 3.3 SECURITY ANALYSIS

We prove that FEVERLESS provides label and data privacy against an adversary controlling at most $n - 2$ clients in the *semi-honest* setting (Smart (2016)). Here, we provide a brief summary of our analysis and theorems. The formal proofs, in the random oracle model, are given in Appendix E.

**Label Privacy:** Label privacy implies that the owner of a label among honest parties should not be leaked to the adversary. We achieve this by using a secure aggregation mechanism where the masks are created via DH key exchange and KDF. In brief, we show that because of the Decisional DH problem (see Definition 4), the adversary cannot distinguish the individual values from randomly chosen ones. That is why the adversary $\mathcal{A}$ cannot learn the owner of the label.

**Data Privacy:** FEVERLESS provides data privacy, meaning that an adversary $\mathcal{A}$ cannot extract the data of any honest party. Individual data values are not separable from random values because of the secure masking. If the source client is not part of the adversary, no data information is leaked. But we require an additional countermeasure for the case where the source client is part of the adversary because it can collect the summation of the data values. We use differential privacy (Dwork et al. (2006a;b)) to achieve data privacy. Because of the noise added by differential privacy, the adversary cannot learn the individual data of an honest client. Moreover, we select the noise clients by the VRF which ensures that the noise leader cannot be predicted or compromised in advance.

**Theorem 3.1** ($\mathcal{A}$ not including source client)**.** *There exists a* PPT *simulator* Sim *for all* $|\mathcal{C}| := n \geq 3$, $|\mathcal{X}| := f \geq n$, $|\mathcal{Y}| := m \geq 1$, $\bigcup_{c \in \mathcal{C}} \mathcal{X}^{(c)}$, $\bigcup_{c \in \mathcal{C}} \mathcal{Y}^{(c)}$ *and* $\mathcal{A} \subset \mathcal{C}$ *so that* $|\mathcal{A}| \leq n - 2$, *the output of* Sim *is indistinguishable from the output of* REAL : $\text{REAL}_{\mathcal{A}}^{\mathcal{C},\mathcal{X},\mathcal{Y}}(\mathcal{X}^{\mathcal{C}}, \mathcal{Y}^{\mathcal{C}}) \equiv \text{Sim}_{\mathcal{A}}^{\mathcal{C},\mathcal{X},\mathcal{Y}}(\mathcal{X}^{\mathcal{A}}, \mathcal{Y}^{\mathcal{A}})$.

**Theorem 3.2** ($\mathcal{A}$ including source client)**.** *There exists a* PPT *simulator* Sim *for all* $|\mathcal{C}| := n \geq 3$, $|\mathcal{X}| := f \geq n$, $|\mathcal{Y}| := m \geq 1$, $\bigcup_{c \in \mathcal{C}} \mathcal{X}^{(c)}$, $\bigcup_{c \in \mathcal{C}} \mathcal{Y}^{(c)}$ *and* $\mathcal{A} \subset \mathcal{C}$ *so that* $|\mathcal{A}| \leq n - 2$, *the output of* Sim *is indistinguishable from the output of* REAL:$\text{REAL}_{\mathcal{A}}^{\mathcal{C},\mathcal{X},\mathcal{Y}}(\mathcal{X}^{\mathcal{C}}, \mathcal{Y}^{\mathcal{C}}) \equiv \text{Sim}_{\mathcal{A}}^{\mathcal{C},\mathcal{X},\mathcal{Y}}(G, H, \mathcal{X}^{\mathcal{A}}, \mathcal{Y}^{\mathcal{A}})$ *where* $G = \sum_{i \in mIDs} g_i^{(c)} + N(0, (\Delta_g \sigma)^2), H = \sum_{i \in mIDs} h_i^{(c)} + N(0, (\Delta_h \sigma)^2)$.

**Theorem 3.3** (Privacy of the Inputs)**.** *No* $\mathcal{A} \subset \mathcal{C}$ *such that* $|\mathcal{A}| \leq n - 2$ *can retrieve the individual values of the honest clients with probability* $1 - \sum_{i=0}^{\hat{k}} C_h^i C_{n-2-h}^{\hat{k}-i} (P_t)^{\hat{k}} (1 - P_t)^{(n-\hat{k})} \frac{C_{\hat{k}-i}^k}{C_{\hat{k}}^k}$, *where* $h$ *and* $\hat{k}$ *refer to the number of non-colluded clients and the number of clients who have selection score larger than threshold, respectively; and* $P_t$ *is the probability of selection score larger than the threshold.*

## 4 EXPERIMENT

We perform evaluations on accuracy, runtime performance and communication cost, and compare our design with two straightforward secure approaches: one is based on LDP (for accuracy), and the other is built on AHE with GDP (for runtime). These approaches are most-commonly-used components for privacy-preserving FL, and they could be the building blocks for complex mechanisms, e.g., MPC. We note the protocol should intuitively outperform those MPC-based solutions, and one may leverage our source code to make further comparison if interested. In the experiments, the baseline, which is the pure XGBoots algorithm, follows the training process of Figure 1 without using any privacy-preserving tools (steps ❷ - ❺). And LDP does not conduct DH key exchange but each client injects noise into the aggregation of gradients and hessians, while AHE follows Figure 1 except executing DH key exchange. In AHE, each client sends (additive) encrypted messages to source client after step ❺. We here show the performance of the best case where there is only one (non-colluded and randomly selected) client adding noise per round ($k = 1$). For other results (where $k \neq 1$), see Appendix H.2. Note we present the communication cost in Appendix H.5).

## 4.1 EXPERIMENT SETUP

To present comprehensive results on accuracy, we set $\epsilon$ to be: 10, 5, 2 and 1, and $\delta$ is set to $10^{-5}$. In terms of accuracy and runtime, we evaluate different situations by varying the number of clients, the number of trees, and the maximum depth of trees (from 2 to 10). Other parameters regarding to training follow the suggestions in (Chen & Guestrin (2016)) and the *library* [5] of XGBoost. To deliver fair results, we conduct each test for 20 independent trials and then calculate the average.

**Datasets.** We run the experiments on three datasets - Credit Card (Yeh & Lien (2009)), Bank Marketing (Moro et al. (2014)) and Banknote Authentication[6] - for classification tasks. To fairly investigate the model performance in DL-VFL, we make the labels as sparse as possible, and they are uniformly distributed on clients. We give the more details of experiment setup in Appendix G.

## 4.2 EVALUATION ON ACCURACY

In Figure 2, we present a clear picture on the accuracy performance based on the #tree and the maximum depth in $(2, 10^{-5})-$DP. We merge the #client in one tree structure, which means in one bar, and the value is the mean of accuracy when conducting on different numbers. The accuracy of the baseline in credit card (about 0.82) and bank marketing (nearly 0.9) remains unchanged as the #tree and maximum depth increases, while the accuracy in banknote authentication rises from 0.9 to approximately 1.0. To highlight the differences and ensure all results to be displayed clearly, we set the ranges of accuracy as $[0.5, 0.9], [0.5, 1]$ and $[0, 1]$ for the three datasets, respectively. Note the performance based on the #client is given in Appendix H.1. Comparing with the baseline, shown in the top and middle rows of the Figure 2, FEVERLESS and LDP suffer from continuously shrinking accuracy as tree structure becomes complex. This is so because the injected noises are accumulated into the model via the increase of query number. And the accuracy is easily affected by the depth. In the worst case where the #tree and maximum depth are both equal to 10, FEVERLESS decreases 10.37% (resp. 14.98% ), and LDP drops 24.78% (resp.24.59%) in credit card (resp. bank marketing). But on average, FEVERLESS' accuracy only shrinks by around 0.9% (resp. 3.21%), while LDP suffers from estimated 3x (resp. 2x) accuracy loss. The difference in the degree of deterioration mainly comes from how much noise is added for each query. We note the deterioration of FEVERLESS is independent with the #client. Thus, we can maintain great accuracy even for the case where there exists a considerable amount of clients.

Despite the fact that less noise is added in FEVERLESS, we do not predict that the accuracy falls to the same level (around 50%, like randomly guess in binary classification) as LDP in the bottom row of Figure 2. This is so because the model is trained by an extremely small-size dataset, which makes it hard to maintain the robustness but relatively sensitive to noise. If setting a larger $\epsilon$, we may see our advantage more clear. The experiments conducted on banknote authentication dataset with larger $\epsilon$ are given in Appendix H.3

To distinguish the performance between FEVERLESS and LDP more clearly, Figure 3 shows the comparison over different $\epsilon$, when #depth and #tree are set to 10. The performance of the model is decayed as the decrease of $\epsilon$. In the left (resp. middle) of the Figure 3, the averaged accuracy of FEVERLESS falls from 0.7686 to 0.5967 (resp. from 0.8517 to 0.6831), while that of LDP also decreases to 0.5299 (resp. 0.5853). We notice that the highest values of LDP stay at the same level as those of FEVERLESS. This is because, in the case of 2-client training, only one client needs to add the noise in LDP (which is identical to our GDP solution). At last, the worse case can be seen in the right of the Figure 3 due to the weak robustness of the model obtained from the banknote authentication. The results are much far away from the baseline there. But even in this case, FEVERLESS still holds a tiny advantage over LDP.

## 4.3 EVALUATION ON TRAINING TIME

To highlight the runtime complexity, we average the results varying by client number into one tree structure as well. We further set the ranges of time as [0s, 9,500s], [0s, 3,500s] and [0s, 110s] for the datasets to deliver visible results. Note since the banknote dataset contains the least samples, it

---

[5]https://xgboost.readthedocs.io/

[6]https://archive.ics.uci.edu/ml/datasets/banknote+authentication

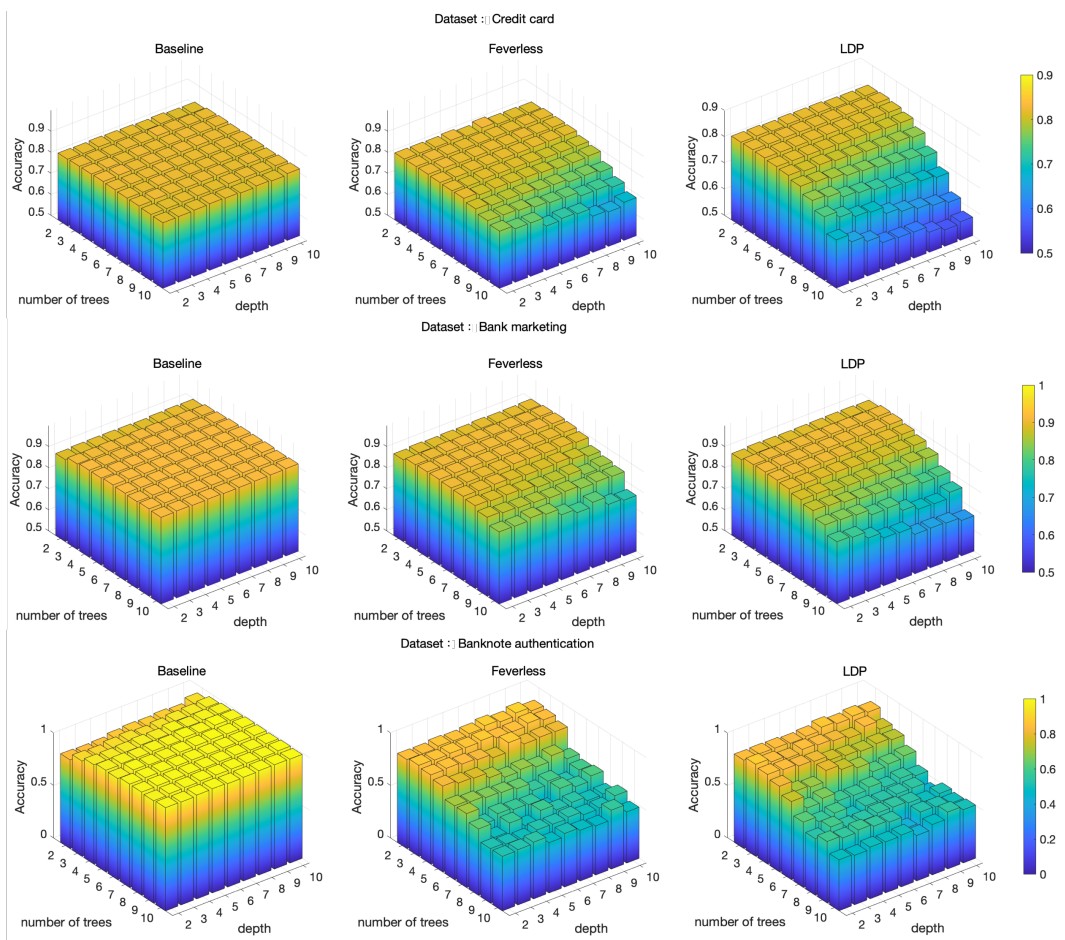

Figure 2: Comparison among the baseline, FEVERLESS and LDP under $\epsilon = 2$. *Top row:* Credit card dataset, accuracy range: [0.5, 0.9]. *Middle row:* Bank marketing, accuracy range: [0.5, 1]. *Bottom row:* Banknote authentication, accuracy range: [0, 1].

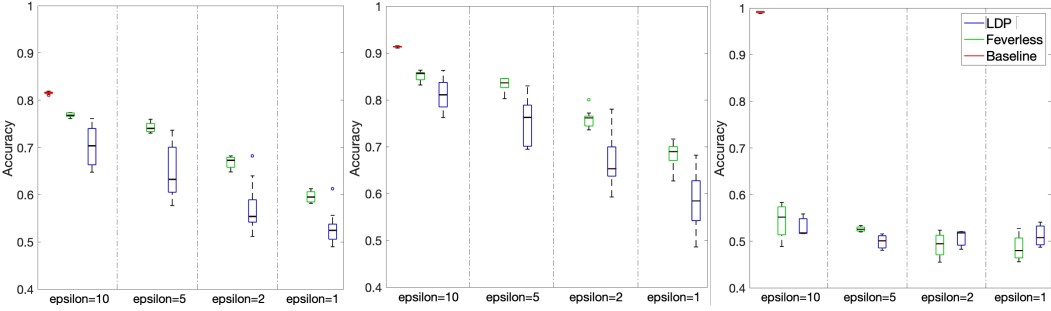

Figure 3: Comparison of accuracy by varying $\epsilon$ in depth=10, the number of trees=10. *Left:* Credit card. *Middle:* Bank marketing. *Right:* Banknote authentication. Accuracy ranges from 0.4 to 1.

does deliver the best training efficiency here. Figure 4 presents the comparison on the training time by varying maximum depth and the number of trees among the datasets.

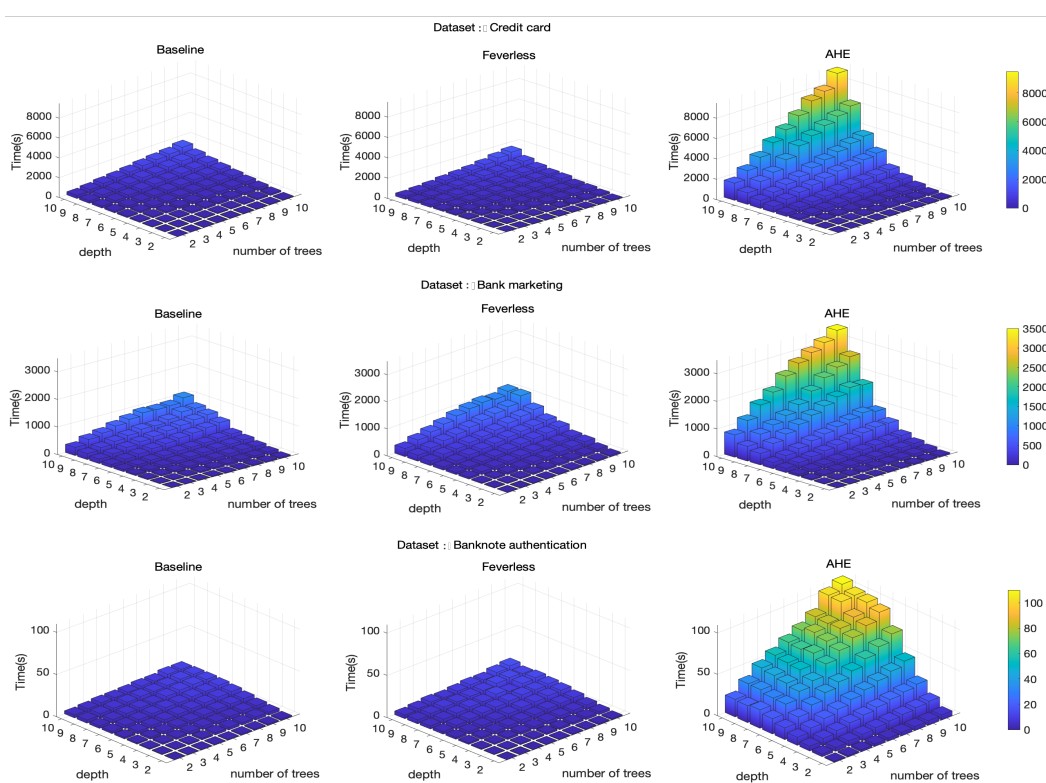

Figure 4: Comparison of time. *Top row:* Credit card dataset, range: [0s, 9,500s]. *Middle row:* Bank marketing, range: [0s, 3,500s]. *Bottom row:* Banknote authentication, range: [0s, 110s].

The training time increases exponentially and linearly with depth and the number of tree, which is consistent with our analysis given in Section 3.2. In Figure 4, compared with the baseline, the runtime of FEVERLESS at most increases 110.3s (resp. 50s, 4.3s), while AHE requires around 70x spike (resp. 48x, 21x) in credit card (resp. bank marketing, banknote authentication), where #depth and #trees are equal to 10. For the average case, FEVERLESS consumes Approx. $1\%(resp.6.5\%, 13.96\%)$ more training time than the baseline, while AHE requires the $351\%(resp.155.1\%, 674\%)$ extra, w.r.t. the three datasets. Its poor performances are due to the laborious calculations in encryption, in which each client has to conduct an encryption per query. By contrast, the masksings in FEVERLESS avoid these excessive costs. We further investigate the runtime performance on the #client in Appendix H.

## 5 CONCLUSION AND FUTURE WORK

We consider a practical scenario where labels are distributedly maintained by different clients for VFL. By leveraging secure aggregation and GDP, we present a novel system, FEVERLESS, to train XGBoost securely. FEVERLESS can achieve perfect secrecy for label and data, and adversaries cannot learn any information about the data if the source client is not corrupted. With DP against differential attack, the source client knows nothing more than summation. Our design is also robust for the collusion of $n-2$ out of n clients. The experiment results show that FEVERLESS is fast and accurate, only taking $1\%$ extra training time, and sacrificing $0.9\%$ accuracy, as compared to the pure XGBoost. In Appendix F, we discuss how to reduce noise, hide distribution of labels and use other security tools. Although our system achieves great performance in terms of security and efficiency, its accuracy still does not work well in small-scale datasets. This remains an open problem. And we will also consider secure solutions against malicious adversaries.

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

## A   NOTATIONS

The frequently used notations are summarized in Table 1.

Table 1: Notations summary

| Notation | Description |
|---|---|
| $\mathcal{X}$ | feature space |
| $X_j^{(c)}$ | $j$-th feature owned by $c$-th client |
| $x_i$ | $i$-th data point with $d$ features |
| $\mathcal{Y}$ | label space |
| $y_i^{(c)}$ | the label of $i$-th data point owned by $c$-th client |
| $\mathcal{I}$ | data index space |
| $\mathcal{C}$ | clients set |
| $g_i^{(c)}$ | the gradient of $i$-th data point owned by $c$-th client |
| $h_i^{(c)}$ | the hessian of $i$-th data point owned by $c$-th client |
| $G$ | summation of gradients |
| $H$ | summation of hessians |
| $m$ | number of data entries |
| $n$ | number of clients |
| $f$ | number of features |
| $d$ | the maximum depth of tree |
| $\epsilon, \delta$ | parameters of differential privacy |
| $\Delta_g$ | sensitivity of gradients |
| $\Delta_h$ | sensitivity of hessians |
| $L_{split}$ | impurity score |
| $w$ | leaf value |
| $pk_c$ | public key generated by $c$-th client |
| $sk_c$ | secret key owned by $c$-th client |
| $g$ | generator of multiplicative group |
| $B_z^j$ | $z$-th bucket of $j$-th feature |

## B   PRELIMINARIES

### B.1   XGBOOST

XGBoost (Chen & Guestrin (2016)) is a popular tree-based model in tabular data training that can provide better interpretation, easier parameters tuning and faster execution speed than deep learning Goodfellow et al. (2016); LeCun et al. (2015). It also outperforms other well-known boosting tree systems in terms of accuracy and efficiency, like Spark MLLib Meng et al. (2016) and H2O Chen & Guestrin (2016), especially for large-scale datasets. Therefore, in this paper, we consider using XGBoost as a building block for classification tasks.

Assume that a training set with $m$ data points composing with feature space $\mathcal{X} = \{x_1, \cdots, x_m\}$ and label space $\mathcal{Y} = \{y_1, \cdots, y_m\}$. Before training starts, every feature will be sorted based their values, and split candidates will be set for features. XGBoost builds trees based on the determination of defined splits candidates and some pruning conditions. Specifically, computing gradients and hessians first according to Eq.(2) and Eq.(3) for each data entry, where $y_i^{(t-1)}$ denotes the prediction of previous tree for $i$-th data point, and $y_i$ is the label of $i$-th data point:

$$g_i = \frac{1}{1 + e^{-y_i^{(t-1)}}} - y_i = \hat{y}_i - y_i, \tag{2}$$

$$h_i = \frac{e^{-y_i^{(t-1)}}}{(1 + e^{-y_i^{(t-1)}})^2}. \tag{3}$$

For splitting nodes, the XGBoost algorithm determines the best split candidate from all others based on maximum $L_{split}$ in Eq.(4), where $\lambda$ and $\gamma$ are regularization parameters:

$$L_{split} = \frac{1}{2}\Big[\frac{\sum_{i\in I_L} g_i}{\sum_{i\in I_L} h_i + \lambda} + \frac{\sum_{i\in I_R} g_i}{\sum_{i\in I_R} h_i + \lambda} - \frac{\sum_{i\in I} g_i}{\sum_{i\in I} h_i + \lambda}\Big] - \gamma. \tag{4}$$

The current node will be the leaf node if the following conditions are fulfilled: reaching the maximum depth of tree, the maximum value of impurity is less than preset threshold. The calculation of the leaf value follows Eq.(5):

$$w = -\frac{\sum_{i\in I} g_i}{\sum_{i\in I} h_i + \lambda}. \tag{5}$$

### B.2 DIFFIE-HELLMAN KEY EXCHANGE

Based on Decision Diffie-Hellman (DDH) hard problem (Boneh (1998)) defined below, Diffie-Hellman key exchange (DH) (Diffie & Hellman (1976)) provides a method used for exchanging keys across public communication channels. Without losing generality and correctness, it consists of a tuple of algorithms (**Param.Gen, Key.Gen, Key.Exc**). The algorithm $(\mathbb{G}, g, q) \leftarrow$ **Param.Gen** $(1^\alpha)$ generates public parameters (a group $\mathbb{G}$ with prime order $q$ generated by a generator $g$) based on secure parameter $\alpha$. $(sk_i, pk_i) \leftarrow$ **Key.Gen**$(\mathbb{G}, g, q)$ allows client $i$ to generate secret key $(sk_i \xleftarrow{\$} \mathbb{Z}_q)$ and compute public key $(pk_i \leftarrow g^{sk_i})$. Shared key is computed by $(pk_i^{sk_j}, pk_j^{sk_i}) \leftarrow$ **Key.Exc**$(sk_i, pk_i, sk_j, pk_j)$. Inspired by (Bonawitz et al. (2017); Ács & Castelluccia (2011)), we utilize shared keys as maskings to protect information of labels against inference attack during transmitting in public channels. The correctness requires $pk_i^{sk_j} = pk_j^{sk_i}$. The security relies on the DDH problem (Boneh (1998)), which is defined as:

**Definition 4** (Decision Diffie-Hellman). *Let $\mathbb{G}$ be a group with prime order $q$ and $g$ be the fixed generator of the group. The Probabilistic Polynomial Time (PPT) adversary $\mathcal{A}$ is given and $g^a$ and $g^b$ where $a$ and $b$ are randomly chosen. The probability of $\mathcal{A}$ distinguishing $(g^a, g^b, g^{ab})$ and $(g^a, g^b, g^c)$ for a randomly chosen $c$ is negligible:*

$$\Big|\Pr\Big[a, b \xleftarrow{\$} \mathbb{Z}_q : \mathcal{A}(g, g^a, g^b, g^{ab}) = \mathsf{true}\Big] -$$
$$\Pr\Big[a, b, c \xleftarrow{\$} \mathbb{Z}_q : \mathcal{A}(g, g^a, g^b, g^c) = \mathsf{true}\Big]\Big| < negl(\alpha).$$

### B.3 PSEUDO-RANDOM GENERATOR AND HASH FUNCTION

Pseudo-Random Generator (PRG) (Håstad et al. (1999)) is an algorithm which is able to generate random numbers. The "pseudo-random" here means that the generated number is not truly random but has the similar properties with random number. Generally, the pseudo-random numbers are determined by given initial values a.k.a seeds. In cryptographic applications, a secure PRG requires attackers not knowing seeds can distinguish a truly random number from a output of PRG with a negligible probability. Similar with PRG, hash function allows mapping arbitrary size of data to a fixed bit value. For reducing communication cost of FEVERLESS, we use `SHAKE-256` (Sha (2015)), one of the hash functions in `SHA-3` (Aumasson et al. (2008)) family, to generate customize size of maskings.

### B.4 KEY DERIVATION FUNCTION

Key Derivation Function (KDF) (Krawczyk & Eronen (2010)) is a kind of hash function that derives multiple secret keys from a main key by utilizing Pesudo-Random Function (PRF) (Kaliski (2005)). In general, KDF algorithm $DK \leftarrow KDF(mainkey, salt, rounds)$ derives keys $DK$ based on a main key, a cryptographic salt and current round of processing algorithm. The security requires a secure KDF is robust for brute-force attack or dictionary attack. Inspired by (Zdziarski (2012)) where key shares generated by DH key exchange are converted to AES keys, in this paper, we use KDF to generate maskings for every round to reduce communication cost. The main key we use is generated by DH key exchange.

### B.5 VERIFIABLE RANDOM FUNCTION

Verifiable Random Function (VRF) (Micali et al. (1999)) is a PRF providing verifiable proofs of correctness of outputs. It is a tool widely used in cryptocurrencies, smart contracts and leader selection in distributed systems (Micali (2016)). Basically, given a input $x$, a signature scheme and a hash function, a practical leader selection scheme with VRF (Micali (2016)) works as:

$$S_{leader} \leftarrow \mathsf{H}(\mathrm{sign}_{sk_i}(x)) \tag{6}$$

where $sk_i$ is the secret key for $i$-th client, and the maximum leader score $S_{leader}$ is used to determine leader. The security and unforgeability of VRF requires that the signature scheme has the property of uniqueness, and hash function is able to map the signature to a random string with fixed size. The correctness of this $S_{leader}$ is proved by the signature of $x$.

### B.6 DIFFERENTIAL PRIVACY

Differential Privacy (DP) (Dwork et al. (2006a;b)) is a data protection system targeting on publishing statistical information of datasets while keeping individual data private. The security of DP requires that adversaries cannot distinguish statistically change from two datasets where an arbitrary data point is different.

The most widely used DP mechanism is called $(\epsilon, \delta)$-DP requiring less noise injected than original proposed $\epsilon$-DP but with the same privacy level. The formal definition is given as follows.

**Definition 5.** $((\epsilon, \delta)$ **- Differential Privacy)** Given two real positive numbers $(\epsilon, \delta)$ and a randomized algorithm $\mathcal{A}: \mathcal{D}^n \to \mathcal{Y}$, the algorithm $\mathcal{A}$ provides $(\epsilon, \delta)$ - differential privacy if for all data sets D, $D^{'} \in \mathcal{D}^n$ differing in only one data sample, and all $\mathcal{S} \subseteq \mathcal{Y}$:

$$Pr[\mathcal{A}(D) \in \mathcal{S}] \leq exp(\epsilon) \cdot Pr[\mathcal{A}(D^{'}) \in \mathcal{S}] + \delta. \tag{7}$$

Note the noise $\mathcal{N} \sim N(0, \Delta^2 \sigma^2)$ will be put into the output of the algorithm, where $\Delta$ is $l_2$ - norm sensitivity of $D$ and $\sigma = \sqrt{2 \ln(1.25/\delta)}$ (Abadi et al. (2016)).

## C PRIVACY CONCERN

Since we assume feature names are not public information for all clients, and the values of features never leave from clients, the privacy issues are mainly incurred by the leakage of label information.

### C.1 INFERENCE ATTACK

During training process, gradients and hessians are sent to source client for $L_{split}$ computation. For classification task, the single gradient is in range $(-1, 0) \cup (0, 1)$ for binary classification. According to Eq.(2), a label can be inferred as 1 and 0 if the range is $(-1, 0)$ and $(0, 1)$, respectively. Besides, hessian illustrated in Eq.(3) can leak a prediction of the corresponding data sample. With training processing, the prediction is increasingly closer to a true label. The source client and outside attackers can infer the true label with high probability. Gradients and hessians cannot be transmitted in plaintext. We thus use secure aggregation scheme to protect them from inference attack.

### C.2 DIFFERENTIAL ATTACK

Differential attack can happen anytime and many times during the calculation of gradients and hessians. Figure 5 describes an example of differential attack taking place in single node split. After sorting feature1, the semi-honest source client defines 2 split candidates and further computes $G_{\{2,5\}} = g_2 + g_5$ and $G_{\{1,2,3,5\}} = g_2 + g_5 + g_1 + g_3$ for the candidates 1 and 2, respectively. Since the source client holds label 2, even if $G_{\{2,5\}}$ is derived by secure aggregation, the $g_5$ still can be revealed by $G_{\{2,5\}} - g_2$.

Another example for differential attack is shown in Figure 6. Assume split candidate 1 is the one for splitting root node. In the current tree structure, source client may split right node by computing $L_{split}$ of split candidate 2. In this case, $G_{\{1,3\}}$ should be aggregated by source client. And the $g_5$ can be revealed by $G_{\{1,2,3,5\}} - G_{\{1,3\}} - g_2$, where $G_{\{1,2,3,5\}}$ is computed in the previous node.

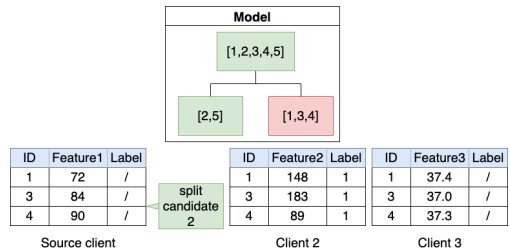

Figure 5: A differential attack on single node split

Figure 6: A differential attack on multiply node splits

# D   MORE DETAILS ON FEVERLESS PROTOCOL

## D.1   XGBOOST TRAINING OVER DISTRIBUTED LABELS

At the initial stage, we allow all clients to agree on a tree structure (maximum depth and the number of trees) and the learning rate for updating prediction. To avoid overfitting problem, we should define regularization parameters. Threshold impurity is also another vital parameter used to identify tree and leaf nodes via the maximum impurity. After that, we should choose $\epsilon$, $\delta$ for DP, hash function for masking generation and noise leader selection. Besides, we select a multiplicative group $\mathbb{G}$ with order $q$ generated by a generator $g$ and a large prime number $p$ to run DH.

At initialization process, all clients set parameters and sort their own feature based on values. Then, split candidates can be defined, and data samples between two different candidates will be grouped as a bucket. At the end, all entries are assigned initialized values to calculate the derivatives of loss function. The detailed algorithm is described as follows.

---

**Algorithm 1:** Initialization

---

1  **Set parameters**: all clients agree on the maximum depth of a tree $d$, the number of trees $(NT)$, learning rate $(\eta)$, regularization parameters $(\lambda, \gamma)$, the threshold of $L_{split}$, $\epsilon$, $\delta$, $p$, $g$, selection portion $(p)$ and hash function

2  **for** $c \in [1, n]$ **do**

3      **for** *each feature $j$ owned by $c$* **do**

4          $\texttt{sort}(X_j^{(c)})$

5          define buckets: $B_z^j$

6      **end**

7      set initialized values: $\hat{y}_i^{(c)}$

8  **end**

---

After initialization, all clients can invoke Algorithm 2 to train model collaboratively. The inputs are from feature space consisting of features $X_j^{(c)}$ and labels $y_i^{(c)}$ distributed on different clients, respectively; while the output is a trained XGBoost model that can be used for prediction. Generally, trees are built one by one. And we see from line 4-10 in Algorithm 2 that each client can compute gradients and hessians at beginning of a new tree construction.

Following that, clients are to split current node. Note that XGBoost training in DL-VFL requires each client to calculate $G$ and $H$. If the labels in some buckets are incomplete, the corresponding

gradients and hessians cannot be computed. Thus, each client should first broadcast missing data index set $mID$ (see line 15-17 in Algorithm 2). Based on the predefined bucket $B_z^j$, $mID$ can be defined if labels in $B_z^j$ are not held by clients. In each broadcast, a client sending messages is regarded as a source client. Then others send the corresponding $g_i^{(c')}$ and $h_i^{(c')}$ back to the source client to compute $L_{split}$ through Algorithm 3-5 depicted in Appendix D.2. After finding a maximum impurity $L_{split\_max}^c$, the current node will be split to "left" and "right" nodes if $L_{split\_max}^c > threshold\_L_{split}$, in which the value of the split candidate is own by $c$. In node splitting, clients should set a given

---

**Algorithm 2:** Protocol overview

1    **Input:** $\{X_j^{(c)} \mid j \in f, c \in |\mathcal{C}|\}$: features, $\{y_i^{(c)} \mid i \in m, c \in |\mathcal{C}|\}$: labels
2    **Output:** XGBoost model
3    **Building trees**:
4    **for** $nt \in [1, NT]$ **do**
5      **for** $c \in [1, n]$ **do**
6        **for** *each data entry $i$ owned by $c$* **do**
7          $g_i^{(c)} \leftarrow \partial_{\hat{y}_i^{(c)}} Loss(\hat{y}_i^{(c)}, y_i^{(c)})$
8          $h_i^{(c)} \leftarrow \partial_{\hat{y}_i^{(c)}}^2 Loss(\hat{y}_i^{(c)}, y_i^{(c)})$
9        **end**
10      **end**
11      **for** *each node in the current tree* **do**
12        **while** *current depth $< d$* **do**
13          **for** $c \in [1, n]$ **do**
14            **for** *each feature $j$ owned by $c$* **do**
15              **for** *each $B_z^j$ owned by $c$* **do**
16                Broadcast $mID = \{i \mid y_i \notin \mathcal{Y}^c\}$
17              **end**
18            aggregate $G$, $H$ by **Algorithm** 3-5
19            compute $L_{split}$ according to Eq.(4)
20          **end**
21          find the maximum $L_{split}^{(c)}$ and broadcast
22          **end**
23        $L_{split\_max}^{(c)} \leftarrow \max(\{L_{split}^{(c)} \mid c \in [1, n]\})$
24        **if** $L_{split\_max}^{(c)} \leq threshold\_L_{split}$ **then**
25          set current node as leaf node
26          $c$ computes $w$ and broadcast
27          Break
28        **else**
29          $c$ splits current node to left node and right node, and broadcasts data index of them.
30        **end**
31      **end**
32      set remaining nodes as leaf nodes
33      $c$ computes $w$ and broadcast
34      clients participating in calculation of $w$: update $\hat{y}_i^{(c)}$
35    **end**
36    **end**

---

node as "leaf" if current depth reaches the predefined maximum depth or the maximum $L_{split}$ is less than the predefined threshold of $L_{split}$ (see line 12, 24-32 in Algorithm 2). The derivation of leaf value is followed by Eq. 5 where $G$ and $H$ are intaken. Since a leaf node is either "left" or "right" split by one of the clients in $\mathcal{C}$ from its parent node, this client knows $G$ and $H$ and leaf value can be derived. Finally, this leaf value will be broadcast, and clients who own the corresponding $g_i^{(c)}$ and $h_i^{(c)}$ can use it to update predictions. The details for the above process are shown in Algorithm 2.

### D.2 Secure Aggregation with Global Differential Privacy

In line 15-19 of Algorithm 2, source client is able to compute $L_{split}$ from the requested missing data indexes and the aggregation of received messages. To avoid that inference and differential attacks are conducted on labels by source client and outside adversaries, we propose a privacy-preserving approach, shown in Algorithm 3-5, to "twist" the DH key exchange, noise leader selection and secure aggregation together. This method represents a viable alternative to train XGBoost securely in DL-VFL without demanding excessive computational resources and affecting model accuracy.

To generate the secure-but-can-be-cancelled-out maskings, we adopt DH here. In Algorithm 3, all clients randomly select numbers as their secret keys and generate the corresponding public keys. For any two clients in the set $\mathcal{C}$, they will exchange public key and compute the corresponding shared keys. For simplicity, we do not describe the signature scheme for DH. We assume DH is conducted on authenticated channels, which means the man-in-the-middle attack (Khader & Lai (2015)) should be invalid here.

---

**Algorithm 3:** Diffie-Hellman key exchange

1 **for** $c \in [1, n]$ **do**
2     $sk_c \leftarrow \mathbb{Z}_p^*$
3 **end**
4 **for** $c \in [1, n]$ **do**
5     $pk_c = g^{sk_c} \bmod p$
6     **for** $c^{'} \in [1, n] \wedge c^{'} \neq c$ **do**
7        $S_{c,c'} = pk_c^{sk_{c'}} \bmod p$
8     **end**
9 **end**

---

If the shared keys are used as maskings directly, our system is not robust for clients collusion unless the amount of communication has been sacrificed as a cost to update maskings per round. But the communication complexity is exponentially increased with the number of clients for a single node splitting. Considering the structure of trees, the overall communication complexity will be $O(2^d \cdot NT \cdot n^2)$, which may not scale well in practical applications.

To tackle this issue, we use KDF to update maskings per round automatically. Specifically, in line 24-25 of Algorithm 5, shared keys are taken as main keys. 0 and 1 are salt values for gradients and hessians, respectively. Since query in each round varies, the generated maskings should be dynamic accordingly. Besides, the sign of maskings is determined by the indexes of clients. In this way, we only need to use DH once, and the communication complexity is independent with tree structure.

To enable FEVERLESS to hold against differential attack, we use GDP approach allowing the chosen one to inject a global noise to aggregated values per round. The approach is quite subtle. If the noise leader is selected by source client, the system will be vulnerable to the collusion. Moreover, a client could be easily identified as a target if we choose it in advance, e.g., selecting a list of leaders before the training. To avoid these issues and limit the probability of collusion to the greatest extent, we use VRF to iteratively select the leader (see Algorithm 4) to securely inject a global noise. The input of VRF includes $mIDs$ and a fresh random number $r$ (line 4 in Algorithm 4), so that this client will not be predicted and set beforehand - reducing its chance to be corrupted in advance by outsiders and the source client.

All clients can broadcast their scores and then the one who holds the "max value" will become the leader. Then the leader re-generates a selection score as score threshold ($selec_{threshold}$) and sends it to the rest of the clients. (line 2-6 in Algorithm 5). The clients send the masked noise back to the leader if the re-generated score is larger than the threshold (line 7-13 in Algorithm 5). Subsequently, the leader will select $\hat{k}$ clients, notify them and aggregate these masked noise to generate a global noise with a random number. In this context, even these selected clients are colluded (note at least one is not) with noise leader and source client, there is still a noise that cannot be recovered, safeguarding the training differentially private. Note since the noise is masked by the random number, the source client (even colluding with the leader) cannot recover the "pure" global

---

**Algorithm 4:** Noise leader selection

---

1   `count` = 1
2   **for** *each time run this algorithm* **do**
3      **for** $c \in [1, n] \ \wedge \ c \neq source \ client$ **do**
4         $selec_c \leftarrow \mathsf{H}(\texttt{SIGN}_{sk_c}(\texttt{count},\texttt{mIDs},\texttt{r}))$
5         `Broadcast`
6      **end**
7      $selec_c^{max} \leftarrow \max(\{selec_c \mid c \in [1, n]\})$
8      set $c$ as noise leader
9      `count+=1`
10   **end**

---

noise to conduct differential attack. And each client adds a noise with a probability $p$. If $k$ out of $\hat{k}$ are non-colluded, the probability of collusion is $(1 - \frac{k}{n})^h$. To cancel out the randomness, the selected clients will subtract the same randomness from masked messages (line 28-31 in Algorithm 5).

Considering that the source client may procrastinate the leader selection and noise injection procedure so as to buy some time for its colluded clients to prepare sufficient large VRF values to participate into the competition of selection and adding noise. One may apply a heartbeat protocol (Nikoletseas & Rolim (2011)) to prevent that a new selected leader intentionally halts the noise adding stage for a long period, say 1 min. If there is no response from the leader after for a short while, a new leader will be randomly selected. Furthermore, the heartbeat may help to solve the problem that the leader accidentally drops from the network. We note that the heartbeat protocol is not our main focus in this paper.

Before replying to source client, we have that the clients with labels put maskings to gradients and hessians, and for those without labels, they just generate and later send out maskings, in which the noise leader (i.e. one of the maskings generators) injects the noise. In this way, the maskings, guaranteeing perfect secrecy of the messages, will be cancelled out after the values aggregation, and the differentially private noise will solidate indistinguishability of individual data entry.

Note that in line 24-34 of Algorithm 5, the maskings and masked values are in the range $[0, N - 1]$. And $N$ should be sufficiently large to avoid overflow, and the summation of gradients and hessians should not exceed $N$.

---

**Algorithm 5:** Secure aggregation with global differential privacy

---

1   **Noise injection:**
2   **if** $c = leader$ **then**
3     $selec_c^{threshold} \leftarrow \mathsf{H}(\mathrm{SIGN}_{sk_c}(\texttt{count},\texttt{mIDs},\texttt{r}))$
4     Broadcast
5     count+=1
6   **end**
7   **for** $c \in [1, n] \ \wedge \ c \neq source\ client \ \wedge \ c \neq noise\ leader$ **do**
8     $selec_c \leftarrow \mathsf{H}(\mathrm{SIGN}_{sk_c}(\texttt{count},\texttt{mIDs},\texttt{r}))$
9     **if** $selec_c > selec_c^{threshold}$ **then**
10       send $\widetilde{n_g^{(c)}} = N(0, \Delta_g^2\sigma^2) + r_g^{(c)}$ and $\widetilde{n_h^{(c)}} = N(0, \Delta_h^2\sigma^2) + r_h^{(c)}$ to noise leader
11       count+=1
12     **end**
13   **end**
14   **if** $c = leader$ **then**
15     $c$ selects $k$ clients from clients of sending noise, $k = \lceil |\{\widetilde{n_g^{(c)}}\}| \cdot p \rceil$
16     **if** $k < 1$ **then**
17       redo noise injection
18     **end**
19     notify $k$ clients
20     noise aggregation: $\widetilde{N_g} = k \cdot N(0, \Delta_g^2\sigma^2) + R_g, \ \widetilde{N_h} = k \cdot N(0, \Delta_h^2\sigma^2) + R_h$
21   **end**
22   **Secure aggregation:**
23   **for** $c \in [1, n]$ **do**
24     $\texttt{mask}_g^{(c)} \leftarrow \left( \sum_{c \neq c'} \frac{|c-c'|}{c-c'} \cdot \left( \mathsf{H}(S_{c,c'}\|0\|\texttt{query}) \bmod N \right) \right) \bmod N$
25     $\texttt{mask}_h^{(c)} \leftarrow \left( \sum_{c \neq c'} \frac{|c-c'|}{c-c'} \cdot \left( \mathsf{H}(S_{c,c'}\|1\|\texttt{query}) \bmod N \right) \right) \bmod N$
26     $G^{(c)} = \sum_{i \in mIDs} g_i^{(c)} + \texttt{mask}_g^{(c)} \bmod N$
27     $H^{(c)} = \sum_{i \in mIDs} h_i^{(c)} + \texttt{mask}_h^{(c)} \bmod N$
28     **if** $selec_c > selec_c^{threshold} \wedge received\ notification$ **then**
29       $G^{(c)} = G^{(c)} - r_g^{(c)} \bmod N$
30       $H^{(c)} = H^{(c)} - r_h^{(c)} \bmod N$
31     **end**
32     **if** $c = leader$ **then**
33       $G^{(c)} = G^{(c)} + \widetilde{N_g} \bmod N$
34       $H^{(c)} = H^{(c)} + \widetilde{N_h} \bmod N$
35     **end**
36     send $\{G^{(c)}, H^{(c)}\}$ to source client
37   **end**

---

# E  SECURITY ANALYSIS

We investigate the security and privacy properties of our protocol. First, we define the security model of our setting and the properties. Then, we prove that our protocol satisfies these properties.

**Security Model.**  Our security is based on the random oracle model (ROM) (Smart (2016)) where the hash function outputs uniformly random value for a new query and the same value for a previously answered query.

**Adversarial Model.**  Our protocol is designed for *semi-honest security* model (Smart (2016)) where all parties follow the protocol while trying to obtain information regarding other parties' inputs. We assume that the source client can collude with other clients, but the size of colluding clients is no more than $n - 2$.

## E.1  PRIVACY GOALS

Our privacy goals can be summarized as:
- *Label privacy*: No adversary controlling at most $n - 2$ clients can learn who is the owner of a label among the honest parties.
- *Data privacy*: No adversary controlling at most $n - 2$ clients can extract the data of an honest party.

We first investigate the case where the source client is not part of the adversary. In the following theorem, we show that there exists a simulator Sim that simulates the joint view of clients in $\mathcal{A}$ by only using the inputs belonging to them. This implies that $\mathcal{A}$ does not learn more than what they have.

**Theorem E.1** ($\mathcal{A}$ not including source client)**.** *There exists a* PPT *simulator* Sim *for all* $|\mathcal{C}| := n \geq 3$, $|\mathcal{X}| := f \geq n$, $|\mathcal{Y}| := m \geq 1$, $\bigcup_{c \in \mathcal{C}} \mathcal{X}^{(c)}$, $\bigcup_{c \in \mathcal{C}} \mathcal{Y}^{(c)}$ *and* $\mathcal{A} \subset \mathcal{C}$ *such that* $|\mathcal{A}| \leq n - 2$, *the output of* Sim *is indistinguishable from the output of* REAL:

$$\text{REAL}_{\mathcal{A}}^{\mathcal{C}, \mathcal{X}, \mathcal{Y}}(\mathcal{X}^{\mathcal{C}}, \mathcal{Y}^{\mathcal{C}}) \equiv \text{Sim}_{\mathcal{A}}^{\mathcal{C}, \mathcal{X}, \mathcal{Y}}(\mathcal{X}^{\mathcal{A}}, \mathcal{Y}^{\mathcal{A}}) \tag{8}$$

*Proof.* In order to prove that simulator Sim can simulate the outputs of the honest parties in $\mathcal{H} := \mathcal{C} - \mathcal{A}$, we show that the distribution of the inputs belonging to the rest of the network cannot be distinguished from a randomly generated data. In this way, the simulator can use any dummy values as inputs of the honest parties to simulate their outputs.

We will simulate the view of the $\mathcal{A}$ regarding the messages broadcast by the honest clients. A client $c$, first makes a key exchange with others, then after some internal operations, outputs $G^{(c)}$ and $H^{(c)}$ values. Let us investigate $G^{(c)}$ value, which is in the form of $\sum_{i \in mIDs} g_i^{(c)} + \text{mask}_g^{(c)}$, except for the noise leader who has additional noise of $N(0, (\Delta_g \sigma)^2)$. The mask values are computed as $\sum_{c \neq c'} \frac{|c - c'|}{c - c'} \cdot \text{H}(S_{c,c'} \| 0 \| \text{query}) \bmod N$.

Here, we will use a hybrid model where we modify the protocol in several steps, and for each step, we will show that modifications are indistinguishable for the adversary $\mathcal{A}$. In the end, we will achieve a hybrid that can be simulated by Sim.

**Hybrid₁**: The first hybrid directly follows the protocol. The distribution of the variables and the view of $\mathcal{A}$ is the same as REAL.

**Hybrid₂**: In the second hybrid, we replace the agreed keys between honest clients $S_{c,c'}$ for all $c, c' \in \mathcal{H}$ with random values $r_{c,c'} \in G$ where $G$ is the group of key exchange protocol $G$. In the original protocol, Diffie-Hellman key exchange is used. The replacement is indistinguishable for the adversary because of the decision Diffie-Hellman assumption given in Definition 4.

Also, note that these random values are only available to parties involved in the key exchange unless they are corrupted by the adversary.

**Hybrid₃**: In this hybrid, we replace the mask values of honest clients $\text{mask}_g^{(c)}$ for all $c \in \mathcal{H}$ with random values $R^{(c)}$. Note that with the replacement in the previous step, the mask values are

computed via $\sum_{c \neq c'} \frac{|c - c'|}{c - c'} \cdot \mathsf{H}(r_{c,c'} \| 0 \| \texttt{query}) \bmod N$ where $r_{c,c'} \in \mathcal{Z}_N$ is a random value that is unknown to the adversary (if both $c$ and $c'$ are honest). Because of the random oracle model, the output of the hash function will be a uniformly random value that is also unknown to the adversary. Since there are at most $n - 2$ clients in $\mathcal{A}$, we have at least two honest clients $c$ and $c'$ for which the adversary cannot know the uniformly chosen output of $\mathsf{H}(r_{c,c'} \| 0 \| \texttt{query})$. Then, the modular summation of these outputs includes at least one value that the adversary does not know and is uniformly random. Thus, it cannot be distinguishable from a random value $R^{(c)}$.

**Hybrid$_4$**: In this hybrid, we replace gradients of honest clients $g_i^{(c)}$ for all $c \in \mathcal{H}$ with '0's. This is done by replacing mask values with $\overline{R^{(c)}} := R^{(c)} - \sum_{i \in mIDs} g_i^{(c)} \bmod N$ to keep the $G^{(c)}$ value the same. From the adversary's perspective, since $R^{(c)}$ values are unknown and uniformly randomly chosen, the replacement is not distinguishable.

In **Hybrid$_4$**, we replace the gradients of honest parties with '0's, and the mask values are replaced by $\overline{R^{(c)}}$ which is unknown to the adversary and chosen from a uniform distribution. Thus, a simulator Sim can simulate the outputs of honest parties $G^{(c)}$ without necessarily knowing their inputs.

The same can be analyzed for hessian value, $H^{(c)}$. Since the masking values of $G^{(c)}$ and $H^{(c)}$ are different and the hash function is modeled as a random oracle, the randomness in both parts of them are independent of each other and indistinguishable by the adversary $\mathcal{A}$. Overall, the simulator Sim can simulate our protocol.

Thus, the view of the $\mathcal{A}$ can be simulated by replacing the inputs of the honest parties with zeros. Thus, the adversary does not learn any information on the inputs of the honest parties. $\qquad\square$

Now, we analyze the case where the source client is part of the $\mathcal{A}$. We show that there exists a simulator Sim that simulates the joint view of clients in $\mathcal{A}$ by only using the inputs belonging to them and the summations $G$ and $H$. This implies that $\mathcal{A}$ does not learn more than what they have and the summation.

**Theorem E.2** ($\mathcal{A}$ including source client). *There exists a* PPT *simulator* Sim *for all* $|\mathcal{C}| := n \geq 3$, $|\mathcal{X}| := f \geq n$, $|\mathcal{Y}| := m \geq 1$, $\bigcup_{c \in \mathcal{C}} \mathcal{X}^{(c)}$, $\bigcup_{c \in \mathcal{C}} \mathcal{Y}^{(c)}$ *and* $\mathcal{A} \subset \mathcal{C}$ *such that* $|\mathcal{A}| \leq n - 2$, *the output of* Sim *is indistinguishable from the output of* REAL*:*

$$\mathrm{REAL}_{\mathcal{A}}^{\mathcal{C}, \mathcal{X}, \mathcal{Y}}(\mathcal{X}^{\mathcal{C}}, \mathcal{Y}^{\mathcal{C}}) \equiv \mathsf{Sim}_{\mathcal{A}}^{\mathcal{C}, \mathcal{X}, \mathcal{Y}}(G, H, \mathcal{X}^{\mathcal{A}}, \mathcal{Y}^{\mathcal{A}}) \tag{9}$$

*where*

$$G = \sum_{i \in mIDs} g_i^{(c)} + N(0, (\Delta_g \sigma)^2), H = \sum_{i \in mIDs} h_i^{(c)} + N(0, (\Delta_h \sigma)^2).$$

*Proof.* Here, we again show that Sim can simulate the outputs of the honest parties in $\mathcal{H}$ without knowing their inputs. Unlike Theorem E.1, Sim is also given the summations $G$ and $H$ because the adversary includes the source client.

We can use the same hybrids with Theorem E.1 until **Hybrid$_4$**, this is because that the inputs of the honest clients are not required yet. We need to update **Hybrid$_4$** such that it takes into account the summation. Here are the hybrids for the $\mathcal{A}$ with source client:

**Hybrid$_1$,Hybrid$_2$,Hybrid$_3$**: The same with Theorem E.1.

**Hybrid$_4$**: In this hybrid, we replace gradients of honest clients $g_i^{(c)}$ for all $c \in \mathcal{H}$ with '0's, except one $c'$ which will be equal to $\sum_{i \in mIDs} g_i^{(\mathcal{H})} \bmod N = G - \sum_{i \in mIDs} g_i^{(\mathcal{A})} \bmod N$. The honest client $c'$ is randomly chosen among $\mathcal{H}$. From the adversary's perspective, since $R^{(c)}$ are unknown uniformly random chosen values, the replacement is not distinguishable.

Overall, the view of the $\mathcal{A}$ can be simulated by replacing the inputs of the honest parties with zeros, except one with $\sum_{i \in mIDs} g_i^{(\mathcal{H})} \bmod N$. Thus, $\mathcal{A}$ does not learn any information from the honest clients, except the summation $\sum_{i \in mIDs} g_i^{(\mathcal{H})} \bmod N$. $\qquad\square$

With Theorem E.2, we show that even the adversary $\mathcal{A}$ with source client cannot know more than the summation of gradient and hessian values, $G$ and $H$. The proof is done via Sim without requiring

individual data of the honest clients except for the summation. This implies that the adversary cannot distinguish which party provided which gradient or hessian values. Moreover, the parties who do not have any of the requested $g$ or $h$ values will send '0' together with the mask (and noise for the leader). This implies that we provide *label privacy*. Meaning that the adversary cannot distinguish which label's $g$ or $h$ values are coming from which honest client.

In the case when the adversary includes the source client, the summation of gradient and hessian values can be known to the adversary. In the following theorem, we show that these summations do not leak any individual data due to differential privacy.

**Theorem E.3** (Privacy of the Inputs). *No $\mathcal{A} \subset \mathcal{C}$ such that $|\mathcal{A}| \leq n - 2$ can retrieve the individual values of the honest clients with probability*

$$1 - \sum_{i=0}^{\hat{k}} C_h^i C_{n-2-h}^{\hat{k}-i} (P_t)^{\hat{k}} (1 - P_t)^{(n-\hat{k})} \frac{C_{\hat{k}-i}^k}{C_{\hat{k}}^k},$$

*where $h$ and $\hat{k}$ refer to the number of non-colluded clients and the number of clients who have selection score larger than threshold. $P_t$ is the probability of selection score larger than the threshold.*

*Proof.* If the adversary does not include the source client, then following the previous theorems, the adversary cannot know any of the inputs belonging to the honest parties. Otherwise, it knows the summations $G$ and $H$. Since we apply differential privacy (Dwork et al. (2006a;b)), the summation cannot leak information regarding the inputs. According to Definition 5, we add differentially private noise guaranteeing the security of individual data points while summation can be calculated.

**Proof of probability.** Note noise leader selects $k$ clients from $n$ clients (rather than itself and the source client) to add noise. Suppose that there are $h$ non-colluded clients out of $n - 2$ clients, the number of clients whose selection scores are larger than the threshold is $\hat{k}$. The number of events is

$$C_{n-2-h}^{\hat{k}} + C_h^1 C_{n-2-h}^{\hat{k}-1} + \cdots + C_h^{\hat{k}} C_{n-2-h}^0,$$

in which the events are that {"there are $\hat{k}$ colluded clients out of $\hat{k}$ clients and 0 non-colluded client",$\cdots$,"there are 0 colluded client out of $\hat{k}$ clients and $\hat{k}$ non-colluded clients"}. Therefore,

$$P(E_i) = C_h^i (P_t)^i (1 - P_t)^{h-i} \cdot C_{n-2-h}^{\hat{k}-i} (P_t)^{\hat{k}-i} (1 - P_t)^{(n-h-\hat{k}+i)}$$
$$= C_h^i C_{n-2-h}^{\hat{k}-i} (P_t)^{\hat{k}} (1 - P_t)^{(n-\hat{k})},$$

where $P_t$ is the probability that the selection score is larger than the threshold, and $E_i$ is $i$-th event. Then, the probability that noise leader selects $k$ colluded clients from $\hat{k}$ clients is $P_0 = \frac{C_{\hat{k}-i}^k}{C_{\hat{k}}^k}$. At the end, the probability of all aggregated noise coming from colluded clients is

$$\sum_{i=0}^{\hat{k}} P(E_i) \cdot P_0 = \sum_{i=0}^{\hat{k}} C_h^i (P_t)^i (1 - P_t)^{h-i} \cdot C_{n-2-h}^{\hat{k}-i} (P_t)^{\hat{k}-i} (1 - P_t)^{(n-h-\hat{k}+i)}$$
$$= \sum_{i=0}^{\hat{k}} C_h^i C_{n-2-h}^{\hat{k}-i} (P_t)^{\hat{k}} (1 - P_t)^{(n-\hat{k})} \frac{C_{\hat{k}-i}^k}{C_{\hat{k}}^k}.$$

Conversely, the probability of at least one non-colluded client participating in noise injection is

$$1 - \sum_{i=0}^{\hat{k}} C_h^i C_{n-2-h}^{\hat{k}-i} (P_t)^{\hat{k}} (1 - P_t)^{(n-\hat{k})} \frac{C_{\hat{k}-i}^k}{C_{\hat{k}}^k}.$$

Note that because of the secure aggregation, the adversary cannot learn anything but the summation. Thus, our protocol does not require the addition of noise to each data. Instead, we only require the noise leader to add the noise, which prevents the retrieval of the individual data from the summation. □

In Theorems E.1 and E.2, we show that $\mathcal{A}$ cannot distinguish the individual values from randomly chosen values and can only know the summation if the source is part of the adversary. In Theorem E.3, we show that $\mathcal{A}$ cannot extract the individual values of the users from the summation due to the added noise and differential privacy. Thus, our protocol satisfies *data privacy*. In other words, the adversary cannot learn the data point of an honest client.

It is important to note that since the noise leader is selected via VRF, no adversary can guess if any honest party will be the leader in the upcoming round beforehand. This provides additional security regarding the manipulation of the noise leader.

## F  DISCUSSION

To reduce the negative impact brought by noise, according to infinity divisibility of Gaussian distribution (Patel & Read (1996)), one may split global noise ($N(0, (\Delta\sigma)^2)$) into $n$ parts ($N(0, \frac{(\Delta\sigma)^2}{n})$). But a drawback is that the privacy budget will increase linearly as an increasing number of colluded clients appear. For example, if GDP achieves $\epsilon$-$DP$, in the worst case where there are $n-1$ colluded clients, the privacy budget will raise to $n \times \epsilon$.

**Hiding labels distribution**. In the semi-honest setting, if the source client sends the missing indexes consistently, adversaries may figure out which labels are distributed (on the source clients) by statistical analysis. We show that this issue can be tackled. In the proposed protocol, source client broadcasts the missing data indexes $mID$ (line 16 of Algorithm 2). Under the semi-honest setting, if source client sends missing indexes consistently, the adversaries will figure out which labels are distributed on source clients by statistic analysis. We note that FEVERLESS can be expanded to avoid this type of leakage by yielding extra communication overheads. Specifically, during broadcasting period, source client should send indexes of one bucket instead of $mID$, and the rest of protocol remains constant. In this way, others cannot distinguish the distribution of labels because all clients share the same index set $\mathcal{I}$. If we assume labels are uniformly distributed on each client, the extra overheads are restricted to $|\mathcal{I}|/|\mathcal{C}|$. This cost is clearly noticeable in those datasets with a large number of data points.

**Other security tools**. The masking scheme realizing secure aggregation may be replaced with an MPC (Damgård et al. (2012); Wu et al. (2020)) or additively homomorphic encryption (Paillier (1999)). However, the major defect of these tools is that they entail labor-intensive calculation with regard to encryption, which may not scale well in large-scale datasets. Due to this concern, we only put light-weight computation in FEVERLESE and further, we enhance the security to "perfect secrecy".

In our design, the selection of noise leader is captured by VRF. We note that there may be other options to fulfil the goal. For example, Proof of Elapsed Time (PoET) (Chen et al. (2017); Corso (2019)) is an interesting and effective mechanism which is used to maintain the consensus of distributed peers in Hyperledger Sawtooth. It provides a fair and trusted lottery strategy to select a block winner (per consensus round). Sharing the same philosophy with the VRF, it may be deployed in our protocol to yield leader. And building a more efficient noise leader selection algorithm could be an interesting open problem.

## G  MORE DETAILS ON EXPERIMENT SETUP

All the experiments are implemented in Python, and conducted on a cluster of machines with Intel(R) Xeon(R) CPU E5-2620 v4 @ 2.10GHz, with 15GB RAM in a local area network.

Intuitively, the smaller $\epsilon$ we set, more secure FEVERLESS will be; but larger noise will be added. We note the above statement can be seen from the experimental results. As for the cryptographic tools, we set the key size of DH and Paillier as 160 bits and 1024 bits respectively(to save some time in running the experiments). This size can reach a symmetric security level with 80 bits key length. Note one may indeed increase the key size to obtain stronger security [7], but this will bring a longer experiment time as a side effect. We use 1024-bit MODP Group with 160-bit Prime Order Subgroup from *RFC 5114* [8] for DH Key exchange. SHAKE-256 (Dworkin (2015)), a member of SHA3 (Dworkin (2015)) family, is used as a hash function in leader selection and secure aggregation.

- **Credit Card:** It is a commercial dataset used for predicting whether costumers will make payment on time. It provides 30,000 samples, and each sample composes of 23 features.
- **Bank marketing:** Consisting with 45,211 data points and 17 features, the goal of bank marketing is to predict if a client will subscribe a term deposit.
- **Banknote authentication:** Offering 1,372 data points and 4 features, this dataset is used to classify authenticated and unauthenticated banknotes. Note that different from traditional tabular data, features in the dataset are extracted from images that are taken from genuine and forged banknote-like specimens through Wavelet Transform (Antonini et al. (1992)). Using the small-scale dataset, the trained model may not be robust for noise, which brings negative impact on accuracy.

## H  ADDITIONAL EXPERIMENTS AND FIGURES

We present additional experiments, and all the experimental settings follow those defined in Section 4.1. In each presented figure, we show the results executed on the datesets Credit card (left), Bank Marketing (middle) and Banknote Authentication (right). Note that the comparison among FEVERLESS, LDP, and AHE requires a condition that #client=2; when #client=1, we can only show the results of the baseline. And the average performance of FEVERLESS in these figures is highlighted as the red dotted line.

Via the experiments, we elaborate that how the accuracy varies with the increasing number of client among the baseline, FEVERLESS and LDP, w.r.t. different tree structures and $\epsilon$. Figure 7-18 are presented for the best case where only a non-colluded client adds the noise. And other cases are demonstrated in Figure 19-26 with the selection scores: 1/2 and 1/3. Beyond those, we also add the comparison results for AHE in Table 2-4 with $\epsilon = 2$.

In general, without any added noise, the baseline can reach the highest accuracy and meanwhile, the accuracy remains stable as the client number increases. The performance of FEVERLESS is right behind that of baseline but still keeps stable. Note there are slight fluctuations in some figures (e.g. Figure 10, 12 and 14), especially for the case where complex tree structure and small $\epsilon$ are used. The LDP approach does harm accuracy, which can be seen from the continuously and significantly falling bars in the figures. Naturally, when more clients engage into the training, more noise should be added into the model. This makes LDP's performance far lower than the red line.

Note that banknote dataset is composed of 4 features. In the VFL setting, every client should have at least one feature. Therefore, we can only allow up to 4 clients to participate in the training. Beside, FEVERLESS does not perform well in banknote dataset. This is so because the model is trained by a small number of samples, so that the robustness is seriously affected by noise.

### H.1  BEST CASE: ACCURACY ON CLIENT NUMBER

---

[7] Note a stronger security level will not affect the training accuracy.
[8] https://tools.ietf.org/html/rfc5114

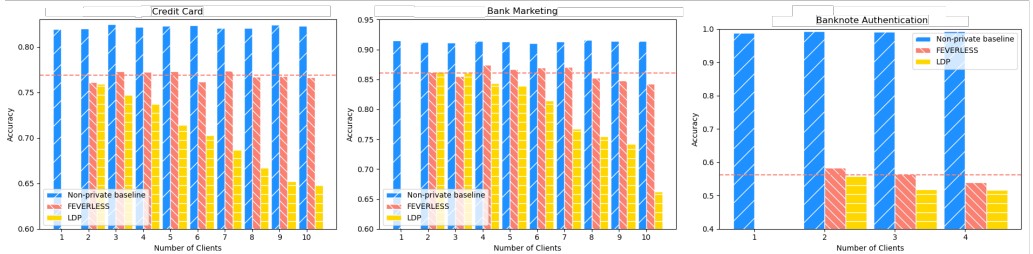

Figure 7: Comparison of accuracy in depth:10, the number of trees:10, epsilon:10.

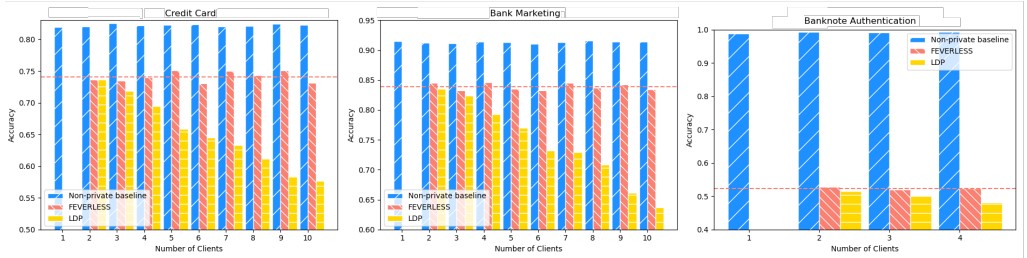

Figure 8: Comparison of accuracy in depth:10, the number of trees:10, epsilon:5.

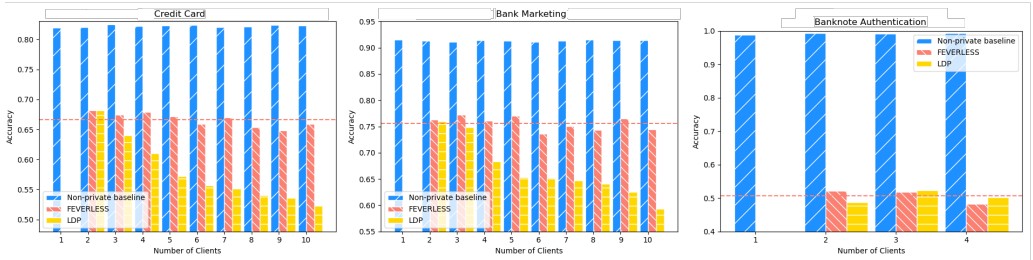

Figure 9: Comparison of accuracy in depth:10, the number of trees:10, epsilon:2.

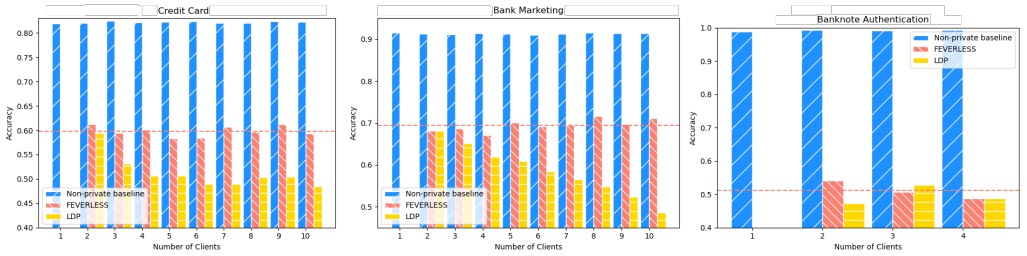

Figure 10: Comparison of accuracy in depth:10, the number of trees:10, epsilon:1.

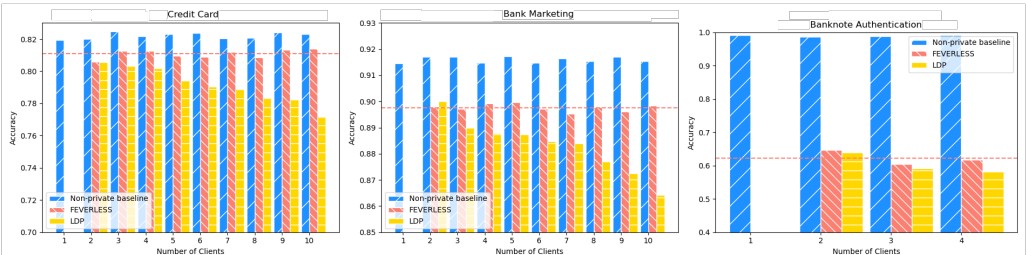

Figure 11: Comparison of accuracy in depth:8, the number of trees:8, epsilon:10.

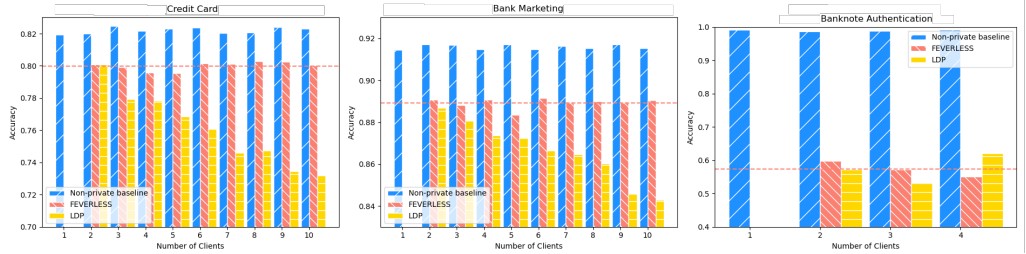

Figure 12: Comparison of accuracy in depth:8, the number of trees:8, epsilon:5.

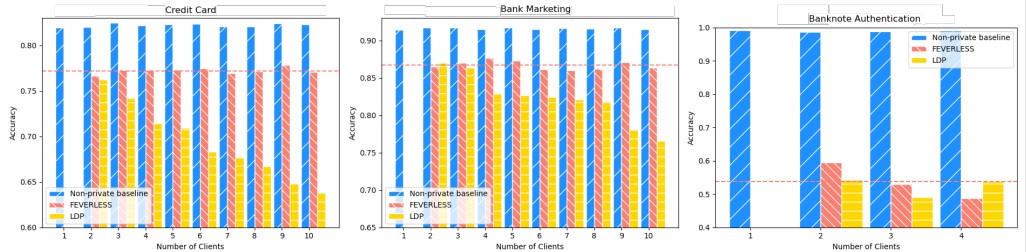

Figure 13: Comparison of accuracy in depth:8, the number of trees:8, epsilon:2.

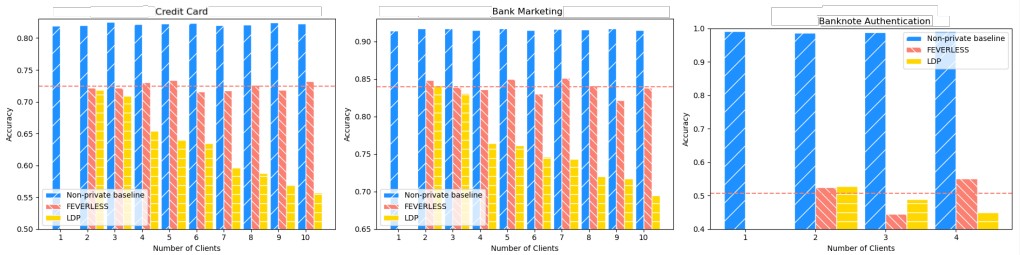

Figure 14: Comparison of accuracy in depth:8, the number of trees:8, epsilon:1.

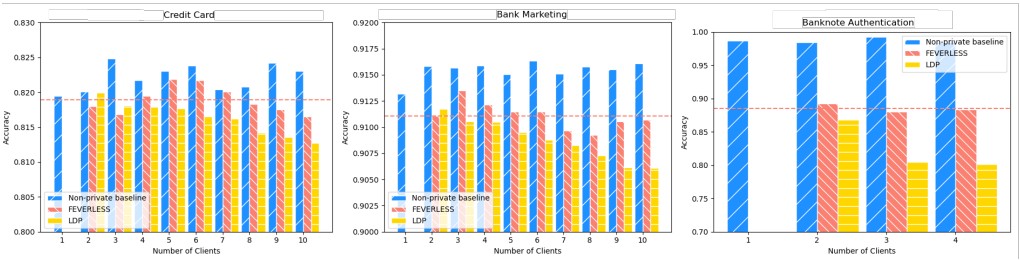

Figure 15: Comparison of accuracy in depth:6, the number of trees:6, epsilon:10.

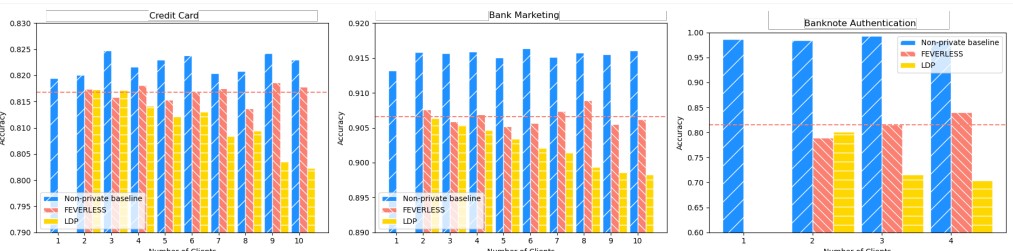

Figure 16: Comparison of accuracy in depth:6, the number of trees:6, epsilon:5.

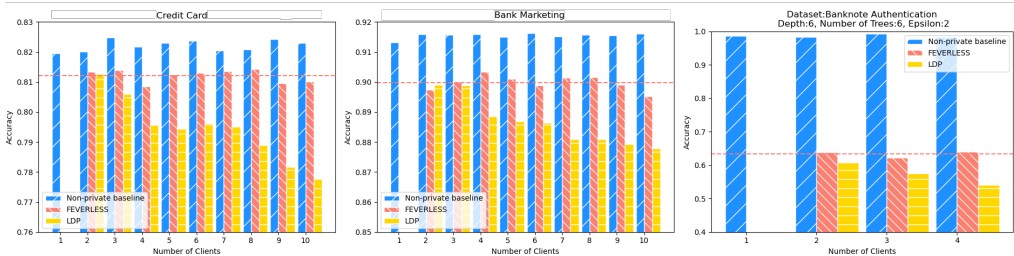

Figure 17: Comparison of accuracy in depth:6, the number of trees:6, epsilon:2.

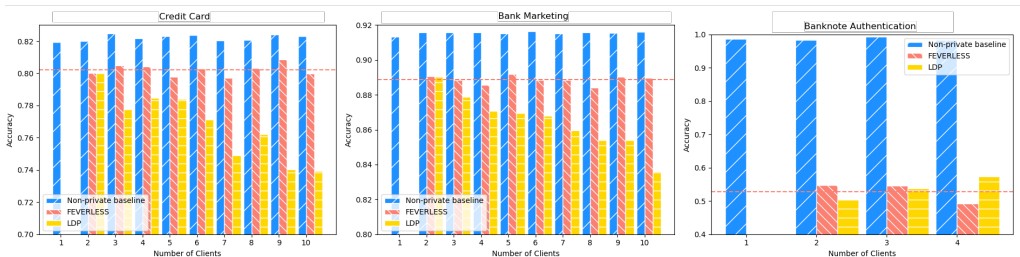

Figure 18: Comparison of accuracy in depth:6, the number of trees:6, epsilon:1.

| Baseline | | | | | |
|---|---|---|---|---|---|
| Accuracy | #tree:2 | #tree:4 | #tree:6 | #tree:8 | #tree:10 |
| Depth:2 | 0.8186 | 0.82 | 0.8175 | 0.8173 | 0.8111 |
| Depth:4 | 0.8219 | 0.8197 | 0.8203 | 0.8226 | 0.8215 |
| Depth:6 | 0.8213 | 0.8204 | 0.8195 | 0.8192 | 0.8206 |
| Depth:8 | 0.8181 | 0.8203 | 0.821 | 0.8163 | 0.816 |
| Depth:10 | 0.8126 | 0.8155 | 0.816 | 0.8171 | 0.8191 |
| **FEVERLESS** | | | | | |
| Accuracy | #tree:2 | #tree:4 | #tree:6 | #tree:8 | #tree:10 |
| Depth:2 | 0.8228 | 0.8185 | 0.8138 | 0.8146 | 0.8119 |
| Depth:4 | 0.8209 | 0.8183 | 0.8201 | 0.8213 | 0.817 |
| Depth:6 | 0.8167 | 0.8155 | 0.8126 | 0.8022 | 0.8009 |
| Depth:8 | 0.8044 | 0.7929 | 0.7795 | 0.7625 | 0.752 |
| Depth:10 | 0.7726 | 0.7547 | 0.7194 | 0.6734 | 0.6821 |
| AHE | | | | | |
| Accuracy | #tree:2 | #tree:4 | #tree:6 | #tree:8 | #tree:10 |
| Depth:2 | 0.8185 | 0.8185 | 0.8225 | 0.8208 | 0.817 |
| Depth:4 | 0.8295 | 0.8195 | 0.8145 | 0.8187 | 0.8178 |
| Depth:6 | 0.8195 | 0.8185 | 0.814 | 0.816 | 0.8053 |
| Depth:8 | 0.807 | 0.7958 | 0.7983 | 0.7748 | 0.7668 |
| Depth:10 | 0.7457 | 0.7168 | 0.719 | 0.7302 | 0.6825 |
| LDP | | | | | |
| Accuracy | #tree:2 | #tree:4 | #tree:6 | #tree:8 | #tree:10 |
| Depth:2 | 0.8195 | 0.8167 | 0.8187 | 0.8146 | 0.8143 |
| Depth:4 | 0.819 | 0.8176 | 0.8173 | 0.8154 | 0.8107 |
| Depth:6 | 0.8119 | 0.8067 | 0.7942 | 0.7842 | 0.7745 |
| Depth:8 | 0.7752 | 0.744 | 0.7259 | 0.7058 | 0.6814 |
| Depth:10 | 0.6843 | 0.6333 | 0.6022 | 0.5772 | 0.5713 |

Table 2: Comparison of accuracy among Baseline, LDP, FEVERLESS and AHE with Credit Card.

| Baseline | | | | | |
|---|---|---|---|---|---|
| Accuracy | #tree:2 | #tree:4 | #tree:6 | #tree:8 | #tree:10 |
| Depth:2 | 0.8887 | 0.893 | 0.8926 | 0.8944 | 0.8962 |
| Depth:4 | 0.9125 | 0.9101 | 0.9136 | 0.9111 | 0.9111 |
| Depth:6 | 0.9168 | 0.9143 | 0.9132 | 0.915 0 | 0.9154 |
| Depth:8 | 0.9118 | 0.9182 | 0.9175 | 0.9145 | 0.9172 |
| Depth:10 | 0.9117 | 0.9115 | 0.9108 | 0.9125 | 0.9152 |
| **FEVERLESS** | | | | | |
| Accuracy | #tree:2 | #tree:4 | #tree:6 | #tree:8 | #tree:10 |
| Depth:2 | 0.8878 | 0.8908 | 0.8927 | 0.8938 | 0.8972 |
| Depth:4 | 0.9051 | 0.9042 | 0.905 | 0.9079 | 0.90588 |
| Depth:6 | 0.9029 | 0.8985 | 0.9032 | 0.8974 | 0.8914 |
| Depth:8 | 0.893 | 0.8869 | 0.8666 | 0.8699 | 0.8509 |
| Depth:10 | 0.8424 | 0.852 | 0.81 | 0.8122 | 0.7654 |
| AHE | | | | | |
| Accuracy | #tree:2 | #tree:4 | #tree:6 | #tree:8 | #tree:10 |
| Depth:2 | 0.887 | 0.8908 | 0.8929 | 0.8962 | 0.8931 |
| Depth:4 | 0.911 | 0.9071 | 0.9026 | 0.9126 | 0.9125 |
| Depth:6 | 0.9025 | 0.8994 | 0.8979 | 0.8995 | 0.9012 |
| Depth:8 | 0.8864 | 0.8885 | 0.8723 | 0.8687 | 0.8541 |
| Depth:10 | 0.8503 | 0.8479 | 0.8229 | 0.8148 | 0.761 |
| LDP | | | | | |
| Accuracy | #tree:2 | #tree:4 | #tree:6 | #tree:8 | #tree:10 |
| Depth:2 | 0.889 | 0.8923 | 0.892 | 0.8939 | 0.8924 |
| Depth:4 | 0.9048 | 0.9054 | 0.9047 | 0.904 | 0.9027 |
| Depth:6 | 0.8983 | 0.8916 | 0.8865 | 0.8816 | 0.8769 |
| Depth:8 | 0.8697 | 0.8523 | 0.8346 | 0.823 | 0.8042 |
| Depth:10 | 0.7951 | 0.7561 | 0.7251 | 0.6891 | 0.6693 |

Table 3: Comparison of accuracy among Baseline, LDP, FEVERLESS and AHE with Bank Marketing.

| Baseline | | | | | |
|---|---|---|---|---|---|
| Accuracy | #tree:2 | #tree:4 | #tree:6 | #tree:8 | #tree:10 |
| Depth:2 | 0.84364 | 0.8444 | 0.8785 | 0.8858 | 0.9229 |
| Depth:4 | 0.9455 | 0.9513 | 0.9607 | 0.9651 | 0.9738 |
| Depth:6 | 0.9629 | 0.9783 | 0.9869 | 0.9935 | 0.992 |
| Depth:8 | 0.9818 | 0.9855 | 0.9847 | 0.9913 | 0.9884 |
| Depth:10 | 0.9818 | 0.9876 | 0.9833 | 0.992 | 0.9884 |
| **FEVERLESS** | | | | | |
| Accuracy | #tree:2 | #tree:4 | #tree:6 | #tree:8 | #tree:10 |
| Depth:2 | 0.8465 | 0.8604 | 0.8655 | 0.864 | 0.8553 |
| Depth:4 | 0.8487 | 0.8378 | 0.7847 | 0.8022 | 0.8058 |
| Depth:6 | 0.7033 | 0.6676 | 0.6378 | 0.5964 | 0.5905 |
| Depth:8 | 0.632 | 0.5738 | 0.5804 | 0.5425 | 0.5084 |
| Depth:10 | 0.5193 | 0.5396 | 0.5084 | 0.4676 | 0.4865 |
| AHE | | | | | |
| Accuracy | #tree:2 | #tree:4 | #tree:6 | #tree:8 | #tree:10 |
| Depth:2 | 0.8473 | 0.8625 | 0.8662 | 0.8716 | 0.859 |
| Depth:4 | 0.9033 | 0.8273 | 0.8058 | 0.7971 | 0.7455 |
| Depth:6 | 0.7251 | 0.6545 | 0.6629 | 0.6436 | 0.5778 |
| Depth:8 | 0.6022 | 0.5611 | 0.5258 | 0.4924 | 0.5255 |
| Depth:10 | 0.4945 | 0.4953 | 0.5015 | 0.4931 | 0.5022 |
| LDP | | | | | |
| Accuracy | #tree:2 | #tree:4 | #tree:6 | #tree:8 | #tree:10 |
| Depth:2 | 0.8415 | 0.8419 | 0.841 | 0.8476 | 0.8439 |
| Depth:4 | 0.8536 | 0.8204 | 0.7634 | 0.7336 | 0.729 |
| Depth:6 | 0.7442 | 0.6606 | 0.5743 | 0.5404 | 0.5343 |
| Depth:8 | 0.5387 | 0.5406 | 0.5549 | 0.5375 | 0.5127 |
| Depth:10 | 0.4848 | 0.4924 | 0.4926 | 0.479 | 0.5072 |

Table 4: Comparison of accuracy among Baseline, LDP, FEVERLESS and AHE with Banknote Authentication.

## H.2  OTHER CASES: ACCURACY ON CLIENT NUMBER

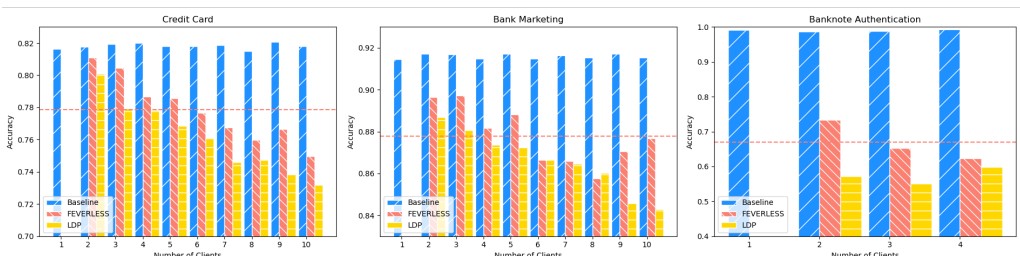

Figure 19: Comparison of accuracy in depth:8, the number of trees:8, epsilon:5, selection score:1/2.

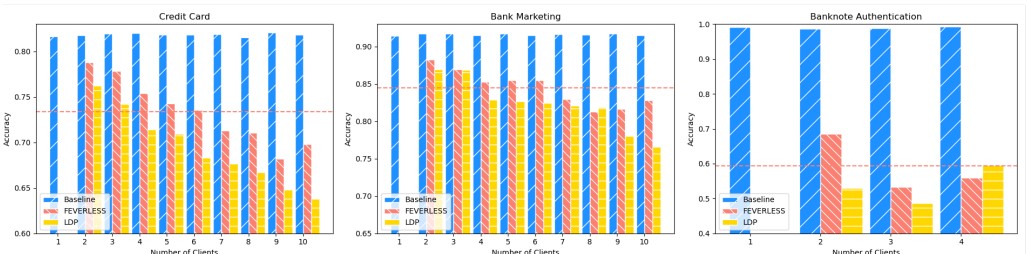

Figure 20: Comparison of accuracy in depth:8, the number of trees:8, epsilon:2, selection score:1/2.

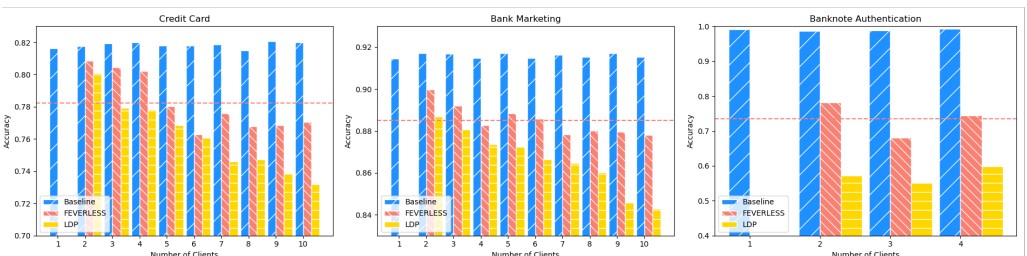

Figure 21: Comparison of accuracy in depth:8, the number of trees:8, epsilon:5, selection score:1/3.

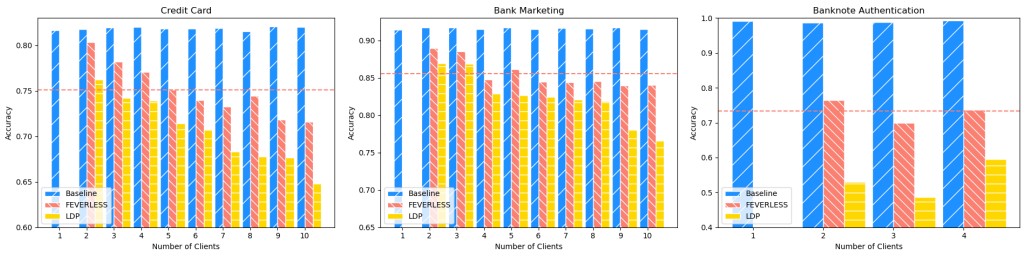

Figure 22: Comparison of accuracy in depth:8, the number of trees:8, epsilon:2, selection score:1/3.

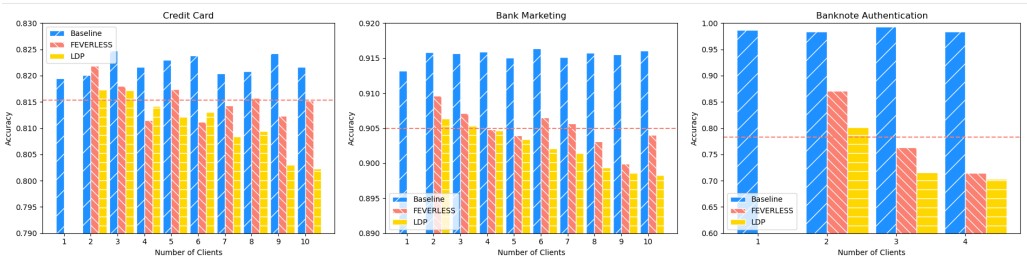

Figure 23: Comparison of accuracy in depth:6, the number of trees:6, epsilon:5, selection score:1/2.

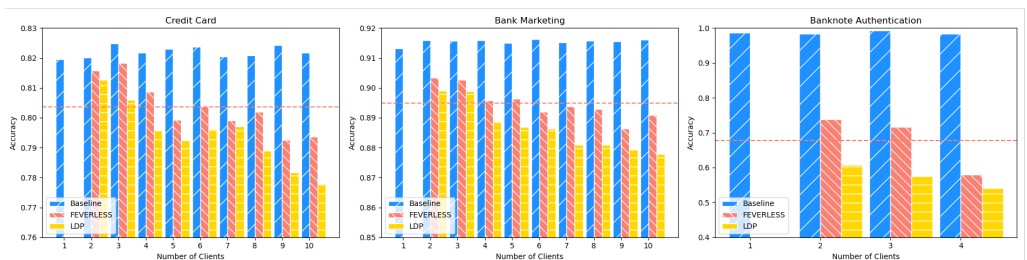

Figure 24: Comparison of accuracy in depth:6, the number of trees:6, epsilon:2, selection score:1/2.

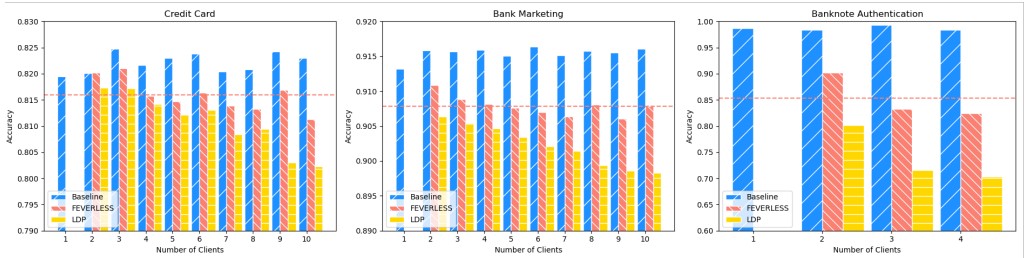

Figure 25: Comparison of accuracy in depth:6, the number of trees:6, epsilon:5, selection score:1/3.

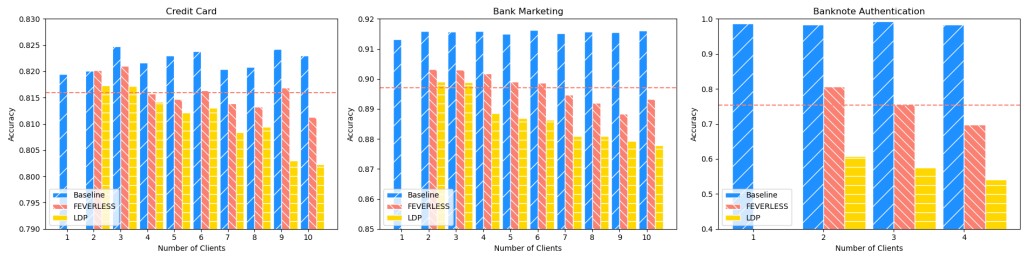

Figure 26: Comparison of accuracy in depth:6, the number of trees:6, epsilon:2, selection score:1/3.

### H.3 ADDITIONAL RESULTS ON ACCURACY FOR BANKNOTE AUTHENTICATION

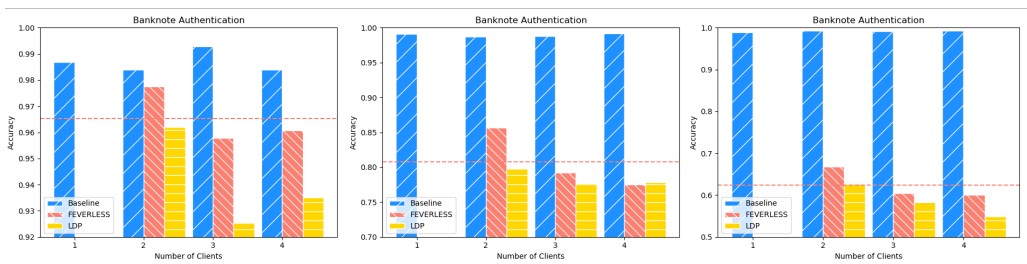

Figure 27: Comparison of accuracy in epsilon:30. *Left:*depth:6, the number of trees:6. *Middle:*depth:8, the number of trees:8. *Right:*depth:10, the number of trees:10

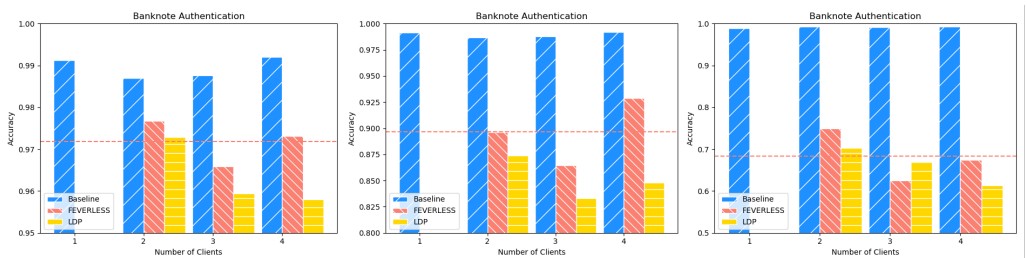

Figure 28: Comparison of accuracy in epsilon:50. *Left:*depth:6, the number of trees:6. *Middle:*depth:8, the number of trees:8. *Right:*depth:10, the number of trees:10

## H.4    ADDITIONAL RESULTS ON TIME

In Figure 29-33, we show the time performance based on various numbers of client, tree and depth. Besides, we present the concrete results in Table 5-7. Table 8 also shows more specific runtime of tree construction in #tree=4 and depth=4 among baseline, FEVERLESS, LDP and AHE. In general, the runtime of FEVERLESS is slightly higher that that of the baseline. Compared to AHE, FEVERLESS significantly reduces training time while preserving privacy. This advantage is clearly seen from the cases using complex tree structures. Note that AHE can be replaced by other more complex cartographic solutions, such as secure MPC, which can also maintain data/label privacy. But the MPC-based solutions will consume more runtime.

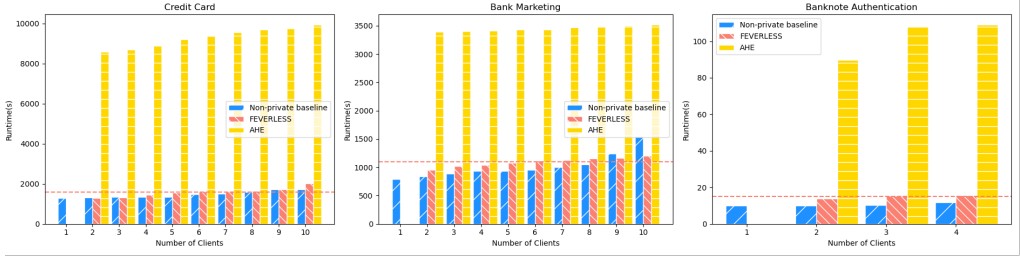

Figure 29: Comparison of runtime in depth:10, the number of trees:10.

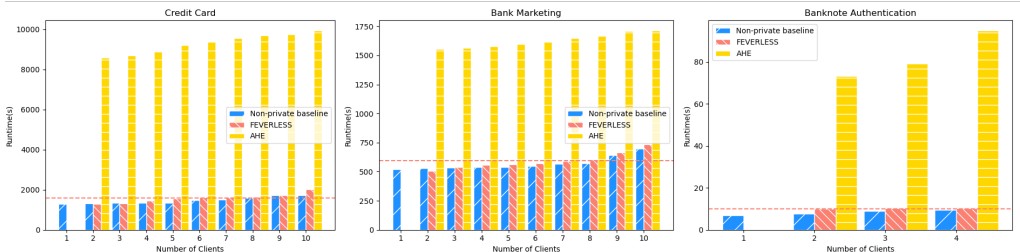

Figure 30: Comparison of runtime in depth:8, the number of trees:8.

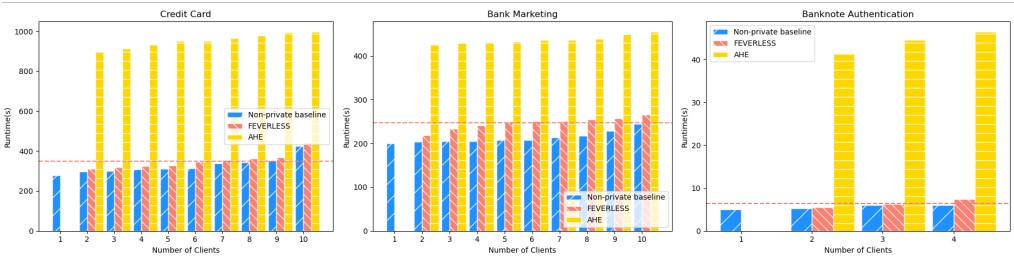

Figure 31: Comparison of runtime in depth:6, the number of trees:6.

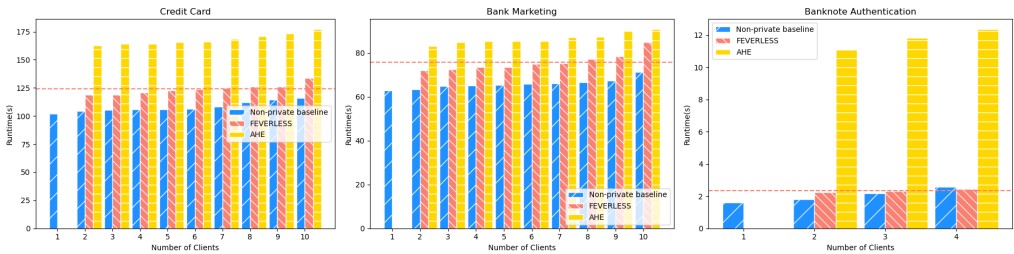

Figure 32: Comparison of runtime in depth:4, the number of trees:4.

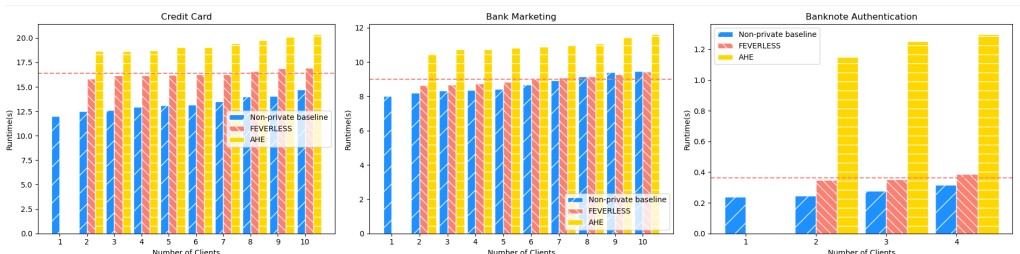

Figure 33: Comparison of runtime in depth:2, the number of trees:2.

| Baseline | | | | | |
|---|---|---|---|---|---|
| Time(s) | #tree:2 | #tree:4 | #tree:6 | #tree:8 | #tree:10 |
| Depth:2 | 13 | 28 | 41 | 56 | 66 |
| Depth:4 | 51 | 109 | 164 | 211 | 259 |
| Depth:6 | 126 | 239 | 332 | 404 | 527 |
| Depth:8 | 197 | 378 | 574 | 710 | 868 |
| Depth:10 | 312 | 591 | 830 | 1144 | 1484 |
| **FEVERLESS** | | | | | |
| Time(s) | #tree:2 | #tree:4 | #tree:6 | #tree:8 | #tree:10 |
| Depth:2 | 16 | 33 | 49 | 65 | 81 |
| Depth:4 | 63 | 124 | 189 | 244 | 303 |
| Depth:6 | 123 | 239 | 350 | 454 | 564 |
| Depth:8 | 258 | 463 | 519 | 687 | 798 |
| Depth:10 | 308 | 518 | 813 | 1129 | 1584 |
| AHE | | | | | |
| Time(s) | #tree:2 | #tree:4 | #tree:6 | #tree:8 | #tree:10 |
| Depth:2 | 19 | 38 | 58 | 77 | 96 |
| Depth:4 | 84 | 168 | 252 | 335 | 459 |
| Depth:6 | 303 | 636 | 954 | 1143 | 1360 |
| Depth:8 | 775 | 1547 | 2366 | 3158 | 3873 |
| Depth:10 | 1685 | 3420 | 5320 | 7360 | 9268 |

Table 5: Comparison of runtime among Baseline, LDP, FEVERLESS and AHE with Credit Card.

| Baseline | | | | | |
|---|---|---|---|---|---|
| Time(s) | #tree:2 | #tree:4 | #tree:6 | #tree:8 | #tree:10 |
| Depth:2 | 9 | 17 | 27 | 37 | 47 |
| Depth:4 | 36 | 76 | 131 | 168 | 222 |
| Depth:6 | 77 | 148 | 247 | 335 | 421 |
| Depth:8 | 164 | 285 | 402 | 575 | 748 |
| Depth:10 | 299 | 576 | 844 | 1007 | 1093 |
| **FEVERLESS** | | | | | |
| Time(s) | #tree:2 | #tree:4 | #tree:6 | #tree:8 | #tree:10 |
| Depth:2 | 9 | 18 | 27 | 37 | 46 |
| Depth:4 | 33 | 66 | 98 | 151 | 161 |
| Depth:6 | 75 | 150 | 215 | 251 | 316 |
| Depth:8 | 160 | 320 | 453 | 593 | 723 |
| Depth:10 | 294 | 548 | 785 | 872 | 1043 |
| AHE | | | | | |
| Time(s) | #tree:2 | #tree:4 | #tree:6 | #tree:8 | #tree:10 |
| Depth:2 | 11 | 22 | 33 | 44 | 56 |
| Depth:4 | 43 | 87 | 130 | 174 | 216 |
| Depth:6 | 147 | 293 | 437 | 600 | 674 |
| Depth:8 | 413 | 840 | 1240 | 1628 | 1773 |
| Depth:10 | 808 | 1613 | 2379 | 2955 | 3481 |

Table 6: Comparison of runtime among Baseline, LDP, FEVERLESS and AHE with Bank Marketing.

| Baseline | | | | | |
|---|---|---|---|---|---|
| Time(s) | #tree:2 | #tree:4 | #tree:6 | #tree:8 | #tree:10 |
| Depth:2 | 0.36 | 0.7 | 0.88 | 1.33 | 1.48 |
| Depth:4 | 1.13 | 2.33 | 3.84 | 3.94 | 4.99 |
| Depth:6 | 1.92 | 3.88 | 5.81 | 7.42 | 9.17 |
| Depth:8 | 2.89 | 5.82 | 6.4 | 10.06 | 11.07 |
| Depth:10 | 3.54 | 6.14 | 7.03 | 8.36 | 10.69 |
| **FEVERLESS** | | | | | |
| Time(s) | #tree:2 | #tree:4 | #tree:6 | #tree:8 | #tree:10 |
| Depth:2 | 0.28 | 0.57 | 0.84 | 1.11 | 1.4 |
| Depth:4 | 1.08 | 2.19 | 3.22 | 4.32 | 5.22 |
| Depth:6 | 2.38 | 4.25 | 6.39 | 8.54 | 10.48 |
| Depth:8 | 3.66 | 6.78 | 8.42 | 8.71 | 13.11 |
| Depth:10 | 4.34 | 7.72 | 8.71 | 10.25 | 14.97 |
| AHE | | | | | |
| Time(s) | #tree:2 | #tree:4 | #tree:6 | #tree:8 | #tree:10 |
| Depth:2 | 1.23 | 2.45 | 3.72 | 4.9 | 6.18 |
| Depth:4 | 5.75 | 11.76 | 17.8 | 23.85 | 29.55 |
| Depth:6 | 14.15 | 29.11 | 44.11 | 59.44 | 76.75 |
| Depth:8 | 22.29 | 41.68 | 65.28 | 82.45 | 101.17 |
| Depth:10 | 21.12 | 46.45 | 64.58 | 82.85 | 102.3 |

Table 7: Comparison of runtime among Baseline, LDP, FEVERLESS and AHE with Banknote Authentication.

| Baseline | | | |
|---|---|---|---|
| Time(s) | Credit Card | Bank Marketing | Banknote Authentication |
| Tree 1 Construction | 29.98 | 17.96 | 0.483 |
| Tree 2 Construction | 28.32 | 17.94 | 0.477 |
| Tree 3 Construction | 26.34 | 17.96 | 0.473 |
| Tree 4 Construction | 25.09 | 17.09 | 0.449 |
| Total | 109.73 | 70.95 | 1.883 |
| LDP | | | |
| Time(s) | Credit Card | Bank Marketing | Banknote Authentication |
| Tree 1 Construction | 28.44 | 19.42 | 0.563 |
| Tree 2 Construction | 26.62 | 18.77 | 0.559 |
| Tree 3 Construction | 27.22 | 18.68 | 0.516 |
| Tree 4 Construction | 28.04 | 19.53 | 0.577 |
| Total | 110.32 | 75.53 | 2.216 |
| **FEVERLESS** | | | |
| Time(s) | Credit Card | Bank Marketing | Banknote Authentication |
| Key exchange | 0.006 | 0.006 | 0.006 |
| Tree 1 Construction | 31.77 | 18.43 | 0.763 |
| Tree 2 Construction | 28.95 | 18.54 | 0.623 |
| Tree 3 Construction | 27.38 | 19.51 | 0.651 |
| Tree 4 Construction | 27.29 | 19.05 | 0.67 |
| Total | 115.39 | 75.53 | 2.713 |
| AHE | | | |
| Time(s) | Credit Card | Bank Marketing | Banknote Authentication |
| Encryption | 9.03 | 6 | 2.37 |
| Tree 1 Construction | 51.62 | 25.65 | 3.91 |
| Tree 2 Construction | 51.03 | 26.2 | 3.84 |
| Tree 3 Construction | 52.17 | 25.78 | 3.93 |
| Tree 4 Construction | 53.55 | 25.01 | 3.92 |
| Total | 217.4 | 108.64 | 17.964 |

Table 8: Comparison of runtime among Baseline, LDP, FEVERLESS and AHE with Credit Card, Bank Marketing and Banknote Authentication. #clients=4, #tree=4, depth=4.

## H.5 RESULTS ON COMMUNICATION COST

In Figure 34-36, we demonstrate the communication cost based on the numbers of clients, tree and depth. For the convenience of comparison, we set #clients=4, #tree=4 and depth=4 as default. We use Table 9-11 to elaborate the concrete costs. To sum up, we see that the communication cost of FEVERLESS is almost the same as those of the baseline and LDP. But as compared to AHE, FEVERLESS significantly reduces costs while maintaining privacy.

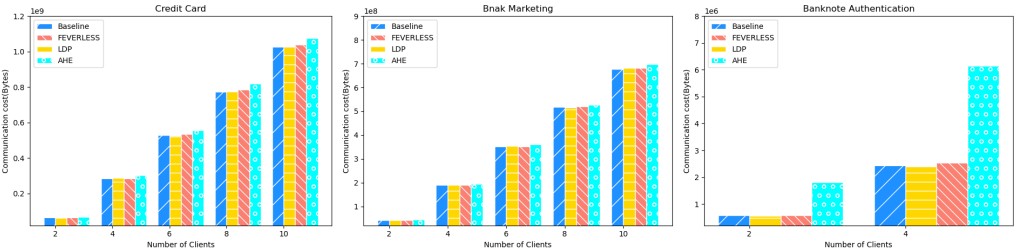

Figure 34: Comparison of communication cost on the number of clients.

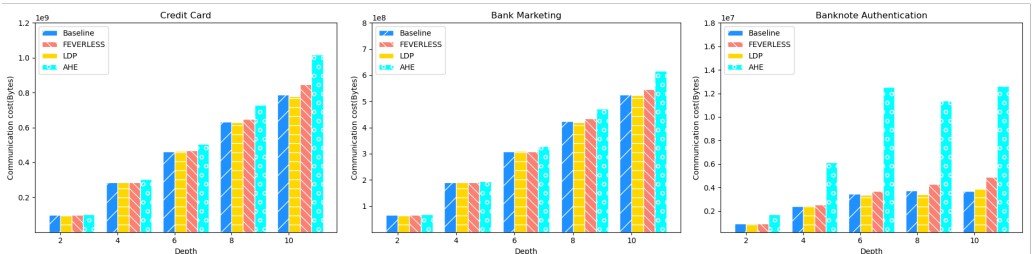

Figure 35: Comparison of communication cost on depth.

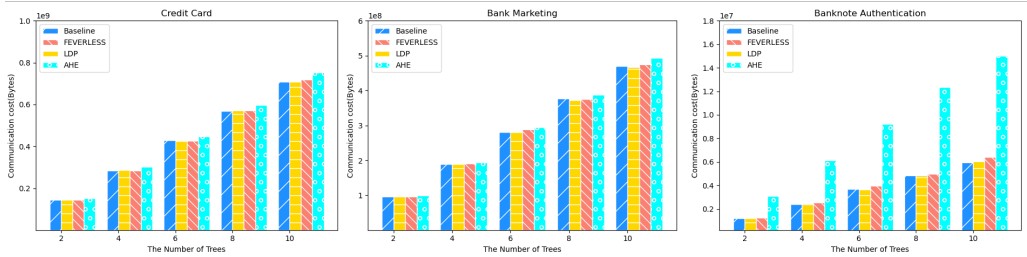

Figure 36: Comparison of communication cost on the number of trees.

| Baseline | | | | | |
|---|---|---|---|---|---|
| Communication cost (byte) | Client 1 | Client 2 | Client 3 | Client 4 | Total |
| Missing IDs | 46318623.75 | 75988029.38 | 80983170 | 79199803.13 | 282489626.3 |
| Key exchange | 0 | 0 | 0 | 0 | 0 |
| Noise leader selection | 0 | 0 | 0 | 0 | 0 |
| Sending noise | 0 | 0 | 0 | 0 | 0 |
| Sending messages | 148824 | 131760 | 127480 | 130352 | 538416 |
| XGBoost update | 460444.375 | 227560.75 | 50247.25 | 209665.75 | 947918.125 |
| Total | 46927892.13 | 76347350.13 | 81160897.25 | 79539820.88 | 283975960.4 |
| LDP | | | | | |
| Communication cost (byte) | Client 1 | Client 2 | Client 3 | Client 4 | Total |
| Missing IDs | 64407678.75 | 82033042.5 | 80595855 | 57890778.75 | 284927355 |
| Key exchange | 0 | 0 | 0 | 0 | 0 |
| Noise leader selection | 0 | 0 | 0 | 0 | 0 |
| Sending noise | 0 | 0 | 0 | 0 | 0 |
| Sending noised messages | 140792 | 127432 | 129544 | 142280 | 540048 |
| XGBoost update | 235180 | 15745.375 | 412955.5 | 284301.625 | 948182.5 |
| Total | 64783650.75 | 82176219.88 | 81138354.5 | 58317360.38 | 286415585.5 |
| FEVERLESS | | | | | |
| Communication cost (byte) | Client 1 | Client 2 | Client 3 | Client 4 | Total |
| Missing IDs | 63060322.5 | 78762268.13 | 62603476.88 | 78498995.63 | 282925063.1 |
| Key exchange | 60 | 60 | 60 | 60 | 240 |
| Noise leader selection | 29092 | 26552 | 29272 | 26732 | 111648 |
| Sending noise | 9556 | 8788 | 9728 | 8920 | 36992 |
| Sending masked messages | 116400 | 106224 | 117112 | 106952 | 446688 |
| XGBoost update | 441000.875 | 53475.5 | 402234.625 | 50431.25 | 947142.25 |
| Total | 63656431.38 | 78957367.63 | 63161883.5 | 78692090.88 | 284467773.4 |
| AHE | | | | | |
| Communication cost (byte) | Client 1 | Client 2 | Client 3 | Client 4 | Total |
| Missing IDs | 63476111.25 | 80963026.88 | 63239242.5 | 80651070 | 288329450.6 |
| Key exchange | 0 | 0 | 0 | 0 | 0 |
| Noise leader selection | 27040 | 25084 | 27408 | 25228 | 104760 |
| Sending noise | 8876 | 8420 | 9028 | 8512 | 34836 |
| Sending encrypted messages | 3462144 | 3211264 | 3508992 | 3229952 | 13412352 |
| XGBoost update | 40627.375 | 229334.75 | 563730.625 | 113990 | 947682.75 |
| Total | 67014798.63 | 84437129.63 | 67348401.13 | 84028752 | 302829081.4 |

Table 9: Comparison of communication cost among Baseline, LDP, FEVERLESS and AHE with Credit Card.

| Baseline | | | | | |
|---|---|---|---|---|---|
| Communication cost (byte) | Client 1 | Client 2 | Client 3 | Client 4 | Total |
| Missing IDs | 92296068 | 64006710 | 15700788 | 15439038 | 187442604 |
| Key exchange | 0 | 0 | 0 | 0 | 0 |
| Noise leader selection | 0 | 0 | 0 | 0 | 0 |
| Sending noise | 0 | 0 | 0 | 0 | 0 |
| Sending messages | 29880 | 44480 | 61456 | 61560 | 197376 |
| XGBoost update | 409814 | 479272 | 299368 | 335316 | 1523770 |
| Total | 92735762 | 64530462 | 16061612 | 15835914 | 189163750 |
| LDP | | | | | |
| Communication cost (byte) | Client 1 | Client 2 | Client 3 | Client 4 | Total |
| Missing IDs | 41687712 | 47518512 | 53953062 | 44605836 | 187765122 |
| Key exchange | 0 | 0 | 0 | 0 | 0 |
| Noise leader selection | 0 | 0 | 0 | 0 | 0 |
| Sending noise | 0 | 0 | 0 | 0 | 0 |
| Sending noised messages | 53248 | 47688 | 46816 | 48760 | 196512 |
| XGBoost update | 826220 | 396448 | 245826 | 53278 | 1521772 |
| Total | 42567180 | 47962648 | 54245704 | 44707874 | 189483406 |
| **FEVERLESS** | | | | | |
| Communication cost (byte) | Client 1 | Client 2 | Client 3 | Client 4 | Total |
| Missing IDs | 48943962 | 42453876 | 26117958 | 71374524 | 188890320 |
| Key exchange | 60 | 60 | 60 | 60 | 240 |
| Noise leader selection | 11720 | 11772 | 14136 | 9856 | 47484 |
| Sending noise | 3840 | 3968 | 4620 | 3080 | 15508 |
| Sending masked messages | 46904 | 47120 | 56560 | 39448 | 190032 |
| XGBoost update | 68820 | 334874 | 456742 | 661876 | 1522312 |
| Total | 49075306 | 42851670 | 26650076 | 72088844 | 190665896 |
| AHE | | | | | |
| Communication cost (byte) | Client 1 | Client 2 | Client 3 | Client 4 | Total |
| Missing IDs | 41995818 | 40671666 | 61365564 | 42890796 | 186923844 |
| Key exchange | 0 | 0 | 0 | 0 | 0 |
| Noise leader selection | 12752 | 12008 | 10656 | 11732 | 47148 |
| Sending noise | 4148 | 3900 | 3580 | 4112 | 15740 |
| Sending encrypted messages | 1632768 | 1537792 | 1364992 | 1502464 | 6038016 |
| XGBoost update | 581494 | 301552 | 264726 | 374816 | 1522588 |
| Total | 44226980 | 42526918 | 63009518 | 44783920 | 194547336 |

Table 10: Comparison of communication cost among Baseline, LDP, FEVERLESS and AHE with Bank Marketing.

| Baseline | | | | | |
|---|---|---|---|---|---|
| Communication cost (byte) | Client 1 | Client 2 | Client 3 | Client 4 | Total |
| Missing IDs | 596475 | 626793.75 | 407826.375 | 606742.125 | 2237837.25 |
| Key exchange | 0 | 0 | 0 | 0 | 0 |
| Noise leader selection | 0 | 0 | 0 | 0 | 0 |
| Sending noise | 0 | 0 | 0 | 0 | 0 |
| Sending messages | 30400 | 29944 | 34504 | 31008 | 125856 |
| XGBoost update | 7496.75 | 128 | 22279.5 | 4403.75 | 34308 |
| Total | 634371.75 | 656865.75 | 464609.875 | 642153.875 | 2398001.25 |
| LDP | | | | | |
| Communication cost (byte) | Client 1 | Client 2 | Client 3 | Client 4 | Total |
| Missing IDs | 441115.125 | 610937.25 | 601907.625 | 619855.5 | 2273815.5 |
| Key exchange | 0 | 0 | 0 | 0 | 0 |
| Noise leader selection | 0 | 0 | 0 | 0 | 0 |
| Sending noise | 0 | 0 | 0 | 0 | 0 |
| Sending noised messages | 33920 | 29984 | 30792 | 29912 | 124608 |
| XGBoost update | 16694.5 | 92 | 10253 | 7273.125 | 34312.625 |
| Total | 491729.625 | 641013.25 | 642952.625 | 657040.625 | 2432736.125 |
| **FEVERLESS** | | | | | |
| Communication cost (byte) | Client 1 | Client 2 | Client 3 | Client 4 | Total |
| Missing IDs | 540828.75 | 608693.25 | 630403.125 | 573725.625 | 2353650.75 |
| Key exchange | 60 | 60 | 60 | 60 | 240 |
| Noise leader selection | 7384 | 6848 | 6852 | 7308 | 28392 |
| Sending noise | 2572 | 2212 | 2220 | 2388 | 9392 |
| Sending masked messages | 29552 | 27424 | 27424 | 29264 | 113664 |
| XGBoost update | 7528.125 | 7366.5 | 8258.5 | 11079.5 | 34232.625 |
| Total | 587924.875 | 652603.75 | 675217.625 | 623825.125 | 2539571.375 |
| AHE | | | | | |
| Communication cost (byte) | Client 1 | Client 2 | Client 3 | Client 4 | Total |
| Missing IDs | 543852.375 | 601610.625 | 602340.75 | 592997.625 | 2340801.375 |
| Key exchange | 0 | 0 | 0 | 0 | 0 |
| Noise leader selection | 7512 | 7200 | 7324 | 7148 | 29184 |
| Sending noise | 2596 | 2360 | 2396 | 2352 | 9704 |
| Sending encrypted messages | 962304 | 922112 | 938240 | 915968 | 3738624 |
| XGBoost update | 14504.625 | 3886.625 | 9163.5 | 6616.5 | 34171.25 |
| Total | 1530769 | 1537169.25 | 1559464.25 | 1525082.125 | 6152484.625 |

Table 11: Comparison of communication cost among Baseline, LDP, FEVERLESS and AHE with Banknote Authentication.

