# OpenReview forum: "FEVERLESS: Fast and Secure Vertical Federated Learning based on XGBoost for Decentralized Labels"
_ICLR.cc/2022/Conference — ICLR 2022 Submitted_

### Official Review · Reviewer_NGrb · 2021-10-30

**Correctness:** 3
**Technical Novelty And Significance:** 2
**Empirical Novelty And Significance:** 2
**Recommendation:** 3
**Confidence:** 3

**Main Review:**

The paper provides a comprehensive security analysis of the proposed protocol. However, I have the following concerns.
1. The paper is not easy to read. The preliminary and many details about the algorithm are put into the appendix, which is not friendly to the readers.
2. The paper provides two examples (i.e., hospitals, bank branches) to demonstrate that the distributed label setting is common in practice. However, these two examples are not convincing. In these two examples, the training examples are usually distributed to the clients (e.g., customers are registered to distinct branches as mentioned in the paper), which is more like a horizontal federated learning setting.
3. In the proposed algorithm, the clients send the sum of the gradients $G$ to the source client. However, the source client needs to know $G_L$ and $G_R$ to compute the gain.  How does the client split the received $G$ into $G_L$ and $G_R$?
4. For a missing label of the source client, there may be multiple clients that hold the label. Then, the noises are injected into the gradients of the same sample. In this case, what is the overall privacy budget for the sample? Also, does the algorithm still work in such as case?
5. The datasets used in the experiments are small. The authors should better conduct experiments on a dataset with at least a million samples.
6. It is not clear to compare the exact performance of different approaches in Figure 2 and Figure 4. The authors can add a table to show the accuracy and time of different approaches with a default depth and number of trees.
7. In the computation cost, the time complexity of generated masks is O(xxx * n), while the overall complexity is O(xxx + n). Is it a typo?


**Summary Of The Paper:**

The paper proposes a new approach called FEVERLESS to conduct vertical federated learning over distributed labels. In the studied setting, the clients have different features of the same sample, and the labels are distributed over different clients. To get the gradients of the samples with missing labels, the clients utilize Diffie-Hellman key exchange, key derivation function, and differential privacy to send the masked gradients to the source client. Then, the source client computes the gain based on the gradients, and the split candidate with the maximum gain is selected as the split point. The authors prove that FEVERLESS protects label and data privacy against an adversary controlling at most n-2 clients. The experiments show that FEVERLESS can achieve comparable accuracy compared with XGBoost.

**Summary Of The Review:**

Overall, I think the paper can be significantly improved in writing. The paper does not convince me about the importance of the distributed label setting. Moreover, I do not fully understand how the algorithm work. Please refer to my main reviews.

---

> ### Author Response · Authors · 2021-11-22
> **Response to Reviewer NGrb**
>
> Thank you for the useful comments. Our response is:
> 1. We've moved security theorems back to the security analysis (please see section 3.3). Due to space limitations, we’ve still kept the crypto background in the appendix, so that we may have more space for the experimental results. But if you insist, we are ok with moving that back to the main body (and meanwhile, putting the experimental results to appendix).
> 2. Sorry for the confusion. We've revised the examples and provided more details, please see Section 1. In the hospital example, feature space is vertically portioned. A cardiac hospital only holds the heart data of the patients, and the mental conditions are managed by a psychiatric center. The authorities may collect and manage their registered patient’s labels locally. Similarly, in the financial example, the banks have part of features on customers (e.g. account balance), while e-commerce companies (e.g., Amazon) have other features (e.g. online transactions). Since these entities use different features on their customers, the data is vertically partitioned.
> 3. “G” is calculated according to the following process. a) The source client determines the split candidates, which can make the IDs of the current node to be divided into the right IDs and the left IDs. b) There is a bucket between (any) two adjacent split candidates, and the source client broadcasts the mIDs in a bucket per round. c) The final aggregation result (G) is the sum of gradients in a bucket. d) After getting the G in each bucket, the source client can calculate GL and GR based on the split candidates. From the above process, we see that G represents the sum of gradients in one bucket rather than all gradients in the current node. Please refer to the Algorithm 2 for the more details.
> 4. Our distributed label assumption is that labels are distributed on different clients, and a label of data instance can only be owned by one client. Therefore, there will not be a situation where the label of a data sample is held by multiple clients at the same time (which could be an interesting open problem). As for the privacy budget, in each round, the noise added by the noise leader strictly follows the definition of global differential privacy, and the source client can only get an aggregated value with differentially private noise. The overall privacy budget will not exceed the pre-set value $\epsilon$.
> 5. Following the state-of-the-art research works based on neural networks, one may easily leverage existing real-world (large-scale) image datasets. But for the context of VFL based on tree structure, the current works, e.g., [1,2,3] commonly use credit card and bank marketing datasets, which are adopted in our experiments. We note that to increase the sample size, [1] did use sklearn to generate "fake data''. We prefer to use real data in the experiments because fake data or simulated data may incur bias in training accuracy. Note that [1] uses fake data for the experiments of computation cost (rather than the accuracy).
> 6. We've added the Table 2-4 in Appendix H.1 and Table 5-7 H.3, respectively.
> 7. We've revised this in Section 3.2.
>
> [1]Yuncheng Wu, Shaofeng Cai, Xiaokui Xiao, Gang Chen, and Beng Chin Ooi.  Privacy-preserving vertical federated learning for tree-based models. Proc. VLDB Endow. , 13(12):2090–2103, July 2020.
>
> [2]Zhihua Tian,  Rui Zhang,  Xiaoyang Hou,  Jian Liu,  and Kui Ren.   Federboost:  Private federated learning for GBDT, 2020
>
> [3] Kewei Cheng, Tao Fan, Yilun Jin, Yang Liu, Tianjian Chen, Dimitrios Papadopoulos, and Qiang Yang. Secureboost: A lossless federated learning framework, 2021.

---

> > ### Comment · Reviewer_NGrb · 2021-11-30
> > **Thanks for your response**
> >
> > Thanks for your response. My major concern is that the distributed label assumption is too strict. It is not practical that a label of data instance can only be owned by one client while labels are distributed on different clients. Even in the provided example, it is not reasonable to assume that each user only loans from a single institution. The setting of the paper still does not convince me. Thus, I'd like to keep my score unchanged.
> >
> > Suggestions:
> > 1. The authors may find some real-world datasets which are exactly the same setting with the assumption (i.e., a label of data instance is owned by one client while the labels are distributed on multiple clients).
> >
> > 2. If the above real-world datasets are not available, the authors may think about how to deal with the case that multiple clients hold the same label. One possible direction is to conduct privacy-preserving record linkage to recognize these labels and assign a single client as the host of the label.
> >
> > 3. If G is the sum of gradients in a bucket, then there should be an index indicating the bucket id. Currently, I only see an index (i.e., $c$ in $G^c$) which indicates the client ID.

---

> > > ### Author Response · Authors · 2021-11-30
> > > **Response to Reviewer NGrb**
> > >
> > > Thanks for your comments. We note that existing works of VFL only consider the scenario where all labels are held by one and only client. In this work, we have extended the scenario and made it much closer to reality. We use the assumption - i.e. a label of a data instance is owned by a client (1-to-1 case) – as a starting point. But our algorithm can still be used for the more practical 1-to-more case. In step 1 of the protocol description (see Section 3.1), the source client will broadcast mIDs to other clients, in which the mIDs include those missing labels. Based on this perspective, no matter how many labels the source client requires, that amount will be reflected on mIDs. For example, in the 1-to-1 (label-client) case, the source client may require more missing labels than the case of 1-to-more.
> > > Our construction is compatible with 1-to-1 and even 1-to-more (label-client) cases. The only difference will rely on the size of mIDs. We note that we will be able to revise the setting description and statements in the paper to reflect the above descriptions.
> > >
> > > In Algorithm 2, the source client traverses all buckets. Since this is the original algorithm of XGBoost, we do not optimize it. We preserve the privacy and correctness on calculating $G$. If we introduce the bucket index, the algorithm will look very complicated - because it involves traversing features, which may not be easily readable to reader. To highlight the secure aggregation algorithm (in order to facilitate understanding), we use $G^{(c)}$ to represent the masked and differentially private values that other clients send to the source client. After the source client aggregates these values, $G$ representing an aggregation of gradients in a single bucket can be generated. We may consider revising Algorithm 2 but this eventually will not affect the correctness and security of training.

---

### Official Review · Reviewer_chCc · 2021-11-02

**Correctness:** 3
**Technical Novelty And Significance:** 2
**Empirical Novelty And Significance:** 3
**Recommendation:** 6
**Confidence:** 2

**Main Review:**

Pros:
1. The problem addressed in this paper is of practical importance for many real-world applications.
2. The challenges and the proposed solutions are well motivated.
3. The paper is also very well-written and has a nice flow.


Cons:
1. Several typos: for example:  “multiply hospitals” should be “multiple”.
2. The main contribution of the paper is to loose the assumption of centralized labels to decentralized labels. Adapting existing secure aggregation to outperform homomorphic encryption-based VFL on latency or differential privacy on accuracy is straightforward. As such, the paper has a limited novelty from the ML perspective.
3. The proposed method has shown improved performance over HE on the aspect of training time and DP on the aspect of accuracy. The paper, however, fails to compare with HE on the aspect of accuracy and DP HE on the aspect of latency.


**Summary Of The Paper:**

This paper studies vertical federated learning for fast and secure XGBoost training where labels are distributed among multiple parties.  Most previous works focus on labels are centralized in one party and adapt cryptography such as homomorphic encryption, multi-party computation, and differential privacy to protect the data and label privacy. This paper instead assumes decentralized labels and combines existing secure aggregation and global differential privacy to safeguard the data and label privacy.

**Summary Of The Review:**

The problem addressed in this paper is of practical importance for many real-world applications. Thus, I would like to see its presentation at the conference.

---

> ### Author Response · Authors · 2021-11-22
> **Response to Reviewer chCc**
>
> Thank you for you useful comments. Here is our response:
> 1. We've done polishing for the paper.
> 2. In terms of privacy-preserving machine learning, the innovation mainly comes from solving security issues that occur during the training process, without seriously sacrificing accuracy and efficiency. Applying secure tools on the machine learning algorithms may not be trivial. This is so because the secure tools have efficiency and security limits in practical use, for example, a fully homomorphic encryption scheme can provide “*” and “+” operations over encrypted gradients, but its efficiency is a serious drawback; and aggregating a considerable amount of clients' noises to hide the final gradient may also seriously harm the training accuracy. In this paper, we propose a more realistic and practical VFL solution to the corresponding privacy, accuracy and efficiency problems.
> 3. Since encryption does not affect the values of gradients, applying AHE will definitely share the same accuracy with FEVERLESS. But the AHE will require more computational resource and communication cost. We've provided extra evidence to prove the statement in Appendix H.1 and H.5. We note that the latency experiments may be more appropriate for the machine learning algorithms with neural networks; and for the VFL context, the runtime among the clients may be sufficient since our model does not require/have in/outcoming neural network operations.

---

### Official Review · Reviewer_dHiP · 2021-11-03

**Correctness:** 2
**Technical Novelty And Significance:** 1
**Empirical Novelty And Significance:** Not applicable
**Recommendation:** 3
**Confidence:** 4

**Main Review:**

1. There are several writing issues with the paper -
i) There should be a background section that introduces the crypto primitives briefly - the main body should be self-contained.
ii) Formal security theorems should be presented in the main paper - only full proofs should be in the Appendix.
iii) A copy-edit pass is required for the paper - some examples
"In practice, a VFL scheme supporting distributed labels
is of necessity." --> is necessary
"without disclosing both feature" --> without disclosing either ... or..
"where a single label is only associated with a client." - client is associated with a single label

2. There is a fundamental flaw in the security guarantee of the paper - the source client and the noise leader could collude, this would reveal the aggregation without noise (the noise can be subtracted out).
3. I could not understand the problem setting - the motivating examples are hospitals and banks. But would not every patient visit a particular hospital or a particular bank (based on locality etc), how is this different from horizontal data split?
4. I could understand why would the info that client A holds label a be made public (Sec 2.3)? Is not label privacy a goal of the scheme?
5. I was confused with the term label space $\mathcal{Y}=\{y_1,\cdots,y_m\}$ - it seems like the label set and not space (label space would be the space of classes like binary for covid test and so on). Also I am confused with the notation $X_j| j \in {1,\cdots, f\}$ since the number of features is $d$ (instead of $f$)?
6. Evaluation is inadequate - performance microbenchmarks such as break down of client/ noise leader and source client time, bandwidth analysis etc are missing.

**Summary Of The Paper:**

The paper proposes a secure protocol in the FL setting for XGBoost where the dataset is vertically split. The proposed mechanism is based on masking and selecting a random client to generate DP noise.

**Summary Of The Review:**

There are several writing issues in the paper. Additionally, the paper suffers from a security flaw.

---

> ### Author Response · Authors · 2021-11-22
> **Response to Reviewer dHiP**
>
> Thank you for the insightful comments. We here give the detailed response one by one.
> 1. i) We agree with you that adding crypto background knowledge in the main body will make the paper more self-contained. Due to the page limit, we’ve put the background to the appendix. If putting it back to the main body, we will not have sufficient space for some experimental contents. Note it will be totally fine with us to keep it in the main body, if you insist.
> ii)We've put security theorems back to the security analysis (please see Section 3.3).
> iii) We've done some polishing for the paper. For example, (a) we've changed "is of necessity" to "is necessary"; (b) Since our privacy-preserving scheme does not just protect features or labels, but does preserve them at the same time, so we keep the expression: "without disclosing both features and labels"; (c) To avoid misunderstanding, we've modified "a single label is only associated with a client" to "a single label belongs to only one client" (please see Section 2.1).
> 2. Thanks for pointing out this. We are sorry that we did not explain sufficient details for the noise leader and noise injection. The previous version showed its best case in noise injection - i.e. only one client ($k$=1) adds the noise, and this client must not be colluded. Accordingly, in Section 4.2 and Appendix H.1, we presented the best-case experimental results. Now, in the revised version, we've added more details related to the adding noise stage, especially when $k$ is >= 1, please see step 5 in Section 3.1 and the Algorithm 5 given in Appendix D.2. And meanwhile, the experimental results related to the our new details have been given in Appendix H.2.
> 3. Sorry for the confusion. We've revised the examples and provided more details, which can be seen in Section 1. In the hospital example, feature space is vertically portioned. A cardiac hospital only holds the heart data of the patients, and the patients’ mental conditions are merely managed by a psychiatric center. Both of the authorities may collect and manage each of its registered patient’s labels locally. Similarly, in the financial example, the banks have part of features on customers (e.g. account balance, withdrawal data), while e-commerce companies (e.g., Amazon) have other features (e.g. online transactions). Since these entities use different features on their customers, the data is vertically partitioned.
> 4. According to the actual privacy needs and definition, we could have two options. The first is to publish the distribution of labels (i.e. enabling one to know which data label belongs to which client) before training, which can save bandwidth because only mIDs are sent out in each round. The second is to hide the distribution of labels, but in each round of training, the source client has to send whole bucket IDs, which will take more communication cost. Note that both of them can effectively prevent the value of the label itself from being breached. In our current version, we use the first option. If one prefers to choose the second option, he/she may just modify the step 1 "broadcast mIDs" of FEVERLESS to "broadcast bucket IDs". The detailed explanation has been given in the “Hiding labels distribution” part of Appendix F.
> 5. We've revised the corresponding notations. Please see Section 2.1.
> 6. We've presented extra evaluations on time and bandwidth, which can be seen in Table 8 in Appendix H.4 and Table 9-11 in Appendix H.5, respectively. Note that in our scenario, each client has part of features and labels. They have the same probability of being the source client and noise leader. Therefore, they are supposed to yield identical runtime. To show the concrete results step by step, we divided total runtime into time of tree construction and key exchange/encryption.

---

### Official Review · Reviewer_Wj2y · 2021-11-07

**Correctness:** 4
**Technical Novelty And Significance:** 3
**Empirical Novelty And Significance:** 3
**Recommendation:** 8
**Confidence:** 3

**Main Review:**

### Strengths

1. The writing and figures are clear.

2. Thorough review of background material.

3. Theoretical and empirical analysis of computation and communication costs demonstrate that the proposed approach compares very favorably to MPC-based solutions.

4. Thorough security analysis proves that the proposed approach is robust to collusion of up to n-2 clients.

5. The proposed combination of Diffie-Hellman key exchange and global differential privacy is ingenious and the experimental results demonstrate the clear advantage of using this combination compared to local differential privacy at each client.

6. The proposed novel setting for vertical federated learning with decentralized labels is very well motivated as shown from the COVID19 example highlighted in the paper.


### Weaknesses

1. The proposed method performs poorly on small datasets.






**Summary Of The Paper:**

The paper introduces a novel setting for vertical federated learning in which labels are distributed among clients, and proposes a novel fast and secure protocol for this setting based on XGBoost . The protocol enables secure aggregation of gradients and hessians for XGBoost via (a) a masking scheme based on Diffie-Hellman key exchange and a key derivation function and (b) global differential privacy. Security analysis presented in the paper demonstrate that label/feature privacy is persevered even if n-2 clients collude. Finally, the efficacy of the proposed protocol is demonstrated via experimental evaluations on multiple datasets.

**Summary Of The Review:**

The paper introduces a novel setting for VFL with decentralized labels, which is very well motivated, and proposes novel protocol for this setting, which is ingenious and shows significant advantages compared to existing methods.

---

> ### Author Response · Authors · 2021-11-22
> **Response to Reviewer Wj2y**
>
> Thanks for the comment. As mentioned in section 4.2, since the models trained with small datasets are quite sensitive to noise, the accuracy falls to about 50%. If we choose to use relatively small noise, our solution (FEVERLESS) will still outperform LDP. To prove this, we've provided extra experimental results on the accuracy of "banknote authentication" using larger epsilons (say 30 and 50), please see Appendix H.4. Note it is unknown that if there is any fine-tuning solution to the problem, i.e. making VFL perform well on small-scale datasets. This could be an interesting open problem.

---

### Author Response · Authors · 2021-11-22
**Summary of main revision**

We appreciate all the reviewers for their efforts and useful comments.
We’ve addressed the main concerns in the response and revised the manuscript according to the comments.
The change we made mainly includes:
1. We’ve given the detailed explanation to the noise injection, and revised the protocol description in Section 3.1, the algorithms in Appendix D and security proof in Appendix E.
2. We’ve added extra experimental results in Appendix H, including:
a) The evaluation on accuracy: Table 2-4 present the concrete results of the accuracy; and Appendix H.2 elaborates other cases.
b) The evaluation on runtime: Table5-7 give the concrete values of runtime; and Table 8 shows the runtime of tree construction.
c) The evaluation on communication cost: Appendix H.5 demonstrates the communication performance based on different metrics.
3. We’ve moved the security theorems back to the main body which can be seen in Section 3.3, and we’ve revised the examples given in the introduction.
4. We’ve revised some typos pointed out by the reviewers. All the modifications are marked in red.

---

### Decision · Program_Chairs · 2022-01-20

**Decision:**

Reject

**Comment:**

The reviewers agree that the problem tackled is important but raise several substantial issues that justify not to accept the paper in its current form. I would encourage the authors to clarify further the crypto part of the paper (dHiP, 2., 4.) and work on how to relax or improve the model assumptions (NGrb). Also, the author's reply to chCc, point 2. becomes more disputable as federated learning is further developed. The argument can be refined.

On a personal note, the statement of Theorem 3.3 could be made clearer, in particular in simplifying (while weakening a bit) the probability bound.

AC.